# Deep Learning without Shortcuts: Shaping the Kernel with Tailored Rectifiers

**Guodong Zhang**[1,2]**, Aleksandar Botev**[3]**, James Martens**[3]
[1]University of Toronto, [2]Vector Institute, [3]DeepMind
gdzhang@cs.toronto.edu, {botev,jamesmartens}@google.com

## Abstract

Training very deep neural networks is still an extremely challenging task. The common solution is to use shortcut connections and normalization layers, which are both crucial ingredients in the popular ResNet architecture. However, there is strong evidence to suggest that ResNets behave more like ensembles of shallower networks than truly deep ones. Recently, it was shown that deep vanilla networks (i.e. networks without normalization layers or shortcut connections) can be trained as fast as ResNets by applying certain transformations to their activation functions. However, this method (called Deep Kernel Shaping) isn't fully compatible with ReLUs, and produces networks that overfit significantly more than ResNets on ImageNet. In this work, we rectify this situation by developing a new type of transformation that is fully compatible with a variant of ReLUs – Leaky ReLUs. We show in experiments that our method, which introduces negligible extra computational cost, achieves validation accuracies with deep vanilla networks that are competitive with ResNets (of the same width/depth), and significantly higher than those obtained with the Edge of Chaos (EOC) method. And unlike with EOC, the validation accuracies we obtain do not get worse with depth.

## 1 Introduction

Thanks to many architectural and algorithmic innovations, the recent decade has witnessed the unprecedented success of deep learning in various high-profile challenges, e.g., the ImageNet recognition task (Krizhevsky et al., 2012), the challenging board game of Go (Silver et al., 2017) and human-like text generation (Brown et al., 2020). Among them, shortcut connections (He et al., 2016a; Srivastava et al., 2015) and normalization layers (Ioffe & Szegedy, 2015; Ba et al., 2016) are two architectural components of modern networks that are critically important for achieving fast training at very high depths, and feature prominently in the ubiquitous ResNet architecture of He et al. (2016b).

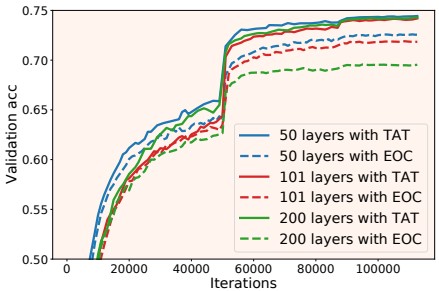

**Figure 1:** Top-1 ImageNet validation accuracy of vanilla deep networks initialized using either EOC (with ReLU) or TAT (with LReLU) and trained with K-FAC.

Despite the success of ResNets, there is significant evidence to suggest that the primary reason they work so well is that they resemble ensembles of shallower networks during training (Veit et al., 2016), which lets them avoid the common pathologies associated with very deep networks (e.g. Hochreiter et al., 2001; Duvenaud et al., 2014). Moreover, ResNets without normalization layers could lose expressivity as the depth goes to infinity (Hayou et al., 2021). In this sense, the question of whether truly deep networks can be efficient and effectively trained on challenging tasks remains an open one.

As argued by Oyedotun et al. (2020) and Ding et al. (2021), the multi-branch topology of ResNets also has certain drawbacks. For example, it is memory-inefficient at inference time, as the input to every residual block has to be kept in memory until the final addition. In particular, the shortcut branches in ResNet-50 account for about 40% of the memory usage by feature maps. Also, the classical interpretation of why deep networks perform well – because of the hierarchical feature representations they produce – does not strictly apply to ResNets, due to their aforementioned tendency to behave like ensembles of shallower networks. Beyond the drawbacks of ResNets, training vanilla deep neural networks (which we define as networks without shortcut connections or

normalization layers) is an interesting research problem in its own right, and finding a solution could open the path to discovering new model architectures. However, recent progress in this direction has not fully succeeded in matching the generalization performance of ResNets.

Schoenholz et al. (2017) used a mean-field analysis of deep MLPs to choose variances for the initial weights and bias parameters, and showed that the resulting method – called Edge of Chaos (EOC) – allowed vanilla networks to be trained at very high depths on small datasets. Building on EOC, and incorporating dynamical isometry theory, Xiao et al. (2018) was able to train vanilla networks with Tanh units[1] at depths of up to 10,000. While impressive, these EOC-initialized networks trained significantly slower than standard ResNets of the same depth, and also exhibited significantly worse generalization performance. Qi et al. (2020) proposed to enforce the convolution kernels to be near isometric, but the gaps with ResNets are still significant on ImageNet. While Oyedotun et al. (2020) was able to narrow the generalization gap between vanilla networks and ResNets, their experiments were limited to networks with only 30 layers, and their networks required many times more parameters. More recently, Martens et al. (2021) introduced a method called Deep Kernel Shaping (DKS) for initializing and transforming networks based on an analysis of their initialization-time kernel properties. They showed that their approach enabled vanilla networks to train faster than previous methods, even matching the speed of similarly sized ResNets when combined with stronger optimizers like K-FAC (Martens & Grosse, 2015) or Shampoo (Anil et al., 2020). However, their method isn't fully compatible with ReLUs, and in their experiments (which focused on training speed) their networks exhibited significantly more overfitting than ResNets.

Inspired by both DKS and the line of work using mean-field theory, we propose a new method called Tailored Activation Transformation (TAT). TAT inherits the main advantages of DKS, while working particularly well with the "Leaky ReLU" activation function. TAT enables very deep vanilla neural networks to be trained on ImageNet without the use of any additional architectural elements, while only introducing negligible extra computational cost. Using TAT, we demonstrate for the first time that a 50-layer vanilla deep network can nearly match the validation accuracy of its ResNet counterpart when trained on ImageNet. And unlike with the EOC method, validation accuracy we achieve does not decrease with depth (see Figure 1). Furthermore, TAT can also be applied to ResNets without normalization layers, allowing them to match or even exceed the validation accuracy of standard ResNets of the same width/depth. A multi-framework open source implementation of DKS and TAT is available at `https://github.com/deepmind/dks`.

## 2 BACKGROUND

Our main tool of analysis will be kernel functions for neural networks (Neal, 1996; Cho & Saul, 2009; Daniely et al., 2016) and the related Q/C maps (Saxe et al., 2013; Poole et al., 2016; Martens et al., 2021). In this section, we introduce our notation and some key concepts used throughout.

### 2.1 KERNEL FUNCTION APPROXIMATION FOR WIDE NETWORKS

For simplicity, we start with the kernel function approximation for feedforward fully-connected networks, and discuss its extensions to convolutional networks and non-feedforward networks later. In particular, we will assume a network that is defined by a sequence of $L$ *combined layers* (each of which is an affine transformation followed by the elementwise activation function $\phi$) as follows:

$$x^{l+1} = \phi\left(W_l x^l + b_l\right) \in \mathbb{R}^{d_{l+1}}, \tag{1}$$

with weights $W_l \in \mathbb{R}^{d_{l+1} \times d_l}$ initialized as $W_l \overset{\text{iid}}{\sim} \mathcal{N}(0, 1/d_l)$ (or scale-corrected uniform orthogonal matrices (Martens et al., 2021)), and biases $b_l \in \mathbb{R}^{d_{l+1}}$ initialized to zero. Due to the randomness of the initial parameters $\theta$, the network can be viewed as random feature model $f_\theta^l(x) \triangleq x^\ell$ at each layer $l$ (with $x^0 = x$) at initialization time. This induces a random kernel defined as follows:

$$\kappa_f^l(x_1, x_2) = f_\theta^l(x_1)^\top f_\theta^l(x_2)/d_l. \tag{2}$$

Given these assumptions, as the width of each layer goes to infinity, $\kappa_f^l(x_1, x_2)$ converges in probability (see Theorem 3) to a deterministic kernel $\tilde{\kappa}_f^l(x_1, x_2)$ that can be computed layer by layer:

$$\Sigma^{l+1} = \mathbb{E}_{z \sim \mathcal{N}(0, \Sigma^l)}\left[\phi(z)\phi(z)^\top\right], \text{ with } \Sigma^l = \begin{bmatrix} \tilde{\kappa}_f^l(x_1, x_1) & \tilde{\kappa}_f^l(x_1, x_2) \\ \tilde{\kappa}_f^l(x_1, x_2) & \tilde{\kappa}_f^l(x_2, x_2) \end{bmatrix}, \tag{3}$$

where $\tilde{\kappa}_f^0(x_1, x_2) = \kappa_f^0(x_1, x_2) = x_1^\top x_2/d_0$.

---

[1]Dynamical isometry is unavailable for ReLU (Pennington et al., 2017), even with orthogonal weights.

## 2.2 LOCAL Q/C MAPS

By equation 3, any diagonal entry $q_i^{l+1}$ of $\Sigma^{l+1}$ only depends on the corresponding diagonal entry $q_i^l$ of $\Sigma^l$. Hence, we obtain the following recursion for these diagonal entries, which we call *q values*:

$$q_i^{l+1} = \mathcal{Q}(q_i^l) = \mathbb{E}_{z \sim \mathcal{N}(0, q_i^l)}[\phi(z)^2] = \mathbb{E}_{z \sim \mathcal{N}(0,1)}\left[\phi(\sqrt{q_i^l}z)^2\right], \text{ with } q_i^0 = \|x_i\|^2/d_0 \quad (4)$$

where $\mathcal{Q}$ is the *local Q map*. We note that $q_i^l$ is an approximation of $\kappa_f^l(x_i, x_i)$. Analogously, one can write the recursion for the normalized off-diagonal entries, which we call *c values*, as:

$$c^{l+1} = \mathcal{C}(c^l, q_1^l, q_2^l) = \frac{\mathbb{E}_{\left[\begin{smallmatrix}z_1\\z_2\end{smallmatrix}\right] \sim \mathcal{N}(0, \Sigma^l)}[\phi(z_1)\phi(z_2)]}{\sqrt{\mathcal{Q}(q_1^l)\mathcal{Q}(q_2^l)}}, \text{ with } \Sigma^l = \begin{bmatrix} q_1^l & \sqrt{q_1^l q_2^l}c^l \\ \sqrt{q_1^l q_2^l}c^l & q_2^l \end{bmatrix}, \quad (5)$$

where $\mathcal{C}$ is the *local C map* and $c^0 = x_1^\top x_2/d_0$. We note that $c^l$ is an approximation of the cosine similarity between $f_\theta^l(x_1)$ and $f_\theta^l(x_2)$. Because $\mathcal{C}$ is a three dimensional function, it is difficult to analyze, as the associated q values can vary wildly for distinct inputs. However, by scaling the inputs to have norm $\sqrt{d_0}$, and rescaling $\phi$ so that $\mathcal{Q}(1) = 1$, it follows that $q_i^l = 1$ for all $l$. This allows us to treat $\mathcal{C}$ as a one dimensional function from $[-1, 1]$ to $[-1, 1]$ satisfying $\mathcal{C}(1) = 1$. Additionally, it can be shown that $\mathcal{C}$ possesses special structure as a *positive definite function* (see Appendix A.4 for details). Going forward, we will thus assume that $q_i^0 = 1$, and that $\phi$ is scaled so that $\mathcal{Q}(1) = 1$.

## 2.3 EXTENSIONS TO CONVOLUTIONAL NETWORKS AND MORE COMPLEX TOPOLOGIES

As argued in Martens et al. (2021), Q/C maps can also be defined for convolutional networks if one adopts a Delta initialization (Balduzzi et al., 2017; Xiao et al., 2018), in which all weights except those in the center of the filter are initialized to zero. Intuitively, this makes convolutional networks behave like a collection of fully-connected networks operating independently over feature map locations. As such, the Q/C map computations for a feed-forward convolutional network are the same as above. Martens et al. (2021) also gives formulas to compute q and c values for weighted sum operations between the outputs of multiple layers (without nonlinearities), thus allowing more complex network topologies. In particular, the sum operation's output q value is given by $q = \sum_{i=1}^n w_i^2 q_i$, and its output c value is given by $\frac{1}{q}\sum_{i=1}^n w_i^2 q_i c_i$. In order to maintain the property that all q values are 1 in the network, we will assume that sum operations are *normalized* in the sense that $\sum_{i=1}^n w_i^2 = 1$.

Following Martens et al. (2021), we will extend the definition of Q/C maps to include *global Q/C maps*, which describe the behavior of entire networks. Global maps, denoted by $\mathcal{Q}_f$ and $\mathcal{C}_f$ for a given network $f$, can be computed by applying the above rules for each layer in $f$. For example, the global C map of a three-layer network $f$ is simply $\mathcal{C}_f(c) = \mathcal{C} \circ \mathcal{C} \circ \mathcal{C}(c)$. Like the local C map, global C maps are positive definite functions (see Appendix A.4). In this work, we restrict our attention to the family of networks comprising of combined layers, and normalized sums between the output of multiple affine layers, for which we can compute global Q/C maps. And all of our formal results will implicitly assume this family of networks.

## 2.4 Q/C MAPS FOR RESCALED RESNETS

ResNets consist of a sequence of residual blocks, each of which computes the sum of a residual branch (which consists of a small multi-layer convolutional network) and a shortcut branch (which copies the block's input). In order to analyze ResNets we will consider the modified version used in Shao et al. (2020) and Martens et al. (2021) which **removes the normalization layers** found in the residual branches, and replaces the sum at the end of each block with a normalized sum. These networks, which we will call *rescaled ResNets*, are defined by the following recursion:

$$x^{l+1} = wx^l + \sqrt{1 - w^2}\mathcal{R}(x^l), \quad (6)$$

where $\mathcal{R}$ is the residual branch, and $w$ is the *shortcut weight* (which must be in $[-1, 1]$). Applying the previously discussed rules for computing Q/C maps, we get $q_i^l = 1$ for all $l$ and

$$c^{l+1} = w^2 c^l + (1 - w^2)\mathcal{C}_\mathcal{R}(c^l). \quad (7)$$

## 3 EXISTING SOLUTIONS AND THEIR LIMITATIONS

Global Q/C maps can be intuitively understood as a way of characterizing signal propagation through the network $f$ at initialization time. The q value approximates the squared magnitude of the activation

vector, so that $\mathcal{Q}_f$ describe the contraction or expansion of this magnitude through the action of $f$. On the other hand, the c value approximates the cosine similarity of the function values for different inputs, so that $\mathcal{C}_f$ describes how well $f$ preserves this cosine similarity from its input to its output.

Standard initializations methods (LeCun et al., 1998; Glorot & Bengio, 2010; He et al., 2015) are motivated through an analysis of how the variance of the activations evolves throughout the network. This can be viewed as a primitive form of Q map analysis, and from that perspective, these methods are trying to ensure that q values remain stable throughout the network by controlling the local Q map. This is necessary for trainability, since very large or tiny q values can cause numerical issues, saturated activation functions (which have implications for C maps), and problems with scale-sensitive losses. However, as was first observed by Schoenholz et al. (2017), a well-behaved C map is also necessary for trainability. When the global C map is close to a constant function (i.e. degenerate) on $(-1, 1)$, which easily happens in deep networks (as discussed in Appendix A.2), this means that the network's output will appear either constant or random looking, and won't convey any useful information about the input. Xiao et al. (2020) and Martens et al. (2021) give more formal arguments for why this leads to slow optimization and/or poor generalization under gradient descent.

Several previous works (Schoenholz et al., 2017; Yang & Schoenholz, 2017; Hayou et al., 2019) attempt to achieve a well-behaved global C map by choosing the variance of the initial weights and biases in each layer such that $\mathcal{C}'(1) = 1$ – a procedure which is referred to as *Edge of Chaos* (EOC). However, this approach only slows down the convergence (with depth) of the c values from exponential to sublinear (Hayou et al., 2019), and does not solve the fundamental issue of degenerate global C maps for very deep networks. In particular, the global C map of a deep network with ReLU and EOC initialization rapidly concentrates around 1 as depth increases (see Figure 2). While EOC allows very deep vanilla networks to be trained, the training speed and generalization performance is typically

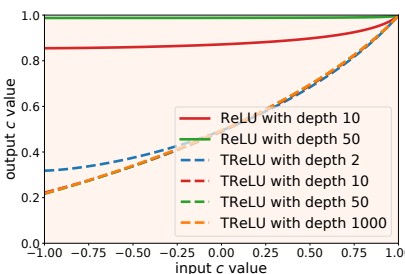

**Figure 2:** Global C maps for ReLU networks (EOC) and TReLU networks ($C_f(0) = 0.5$). The global C map of a TReLU network converges to a well-behavior function as depth increases (proved in Proposition 3).

much worse than for comparable ResNets. Klambauer et al. (2017) applied an affine transformation to the output of activation functions to achieve $\mathcal{Q}(1) = 1$ and $\mathcal{C}(0) = 0$, while Lu et al. (2020) applied them to achieve $\mathcal{Q}(1) = 1$ and $\mathcal{C}'(1) = 1$, although the effect of both approaches is similar to EOC.

To address these problems, Martens et al. (2021) introduced DKS, which enforces the conditions $\mathcal{C}_f(0) = 0$ and $\mathcal{C}'_f(1) = \zeta$ (for some modest constant $\zeta > 1$) directly on the network's global C map $\mathcal{C}_f$. They show that these conditions, along with the positive definiteness of C maps, cause $\mathcal{C}_f$ to be close to the identity and thus well-behaved. In addition to these C map conditions, DKS enforces that $\mathcal{Q}(1) = 1$ and $\mathcal{Q}'(1) = 1$, which lead to constant q values of 1 in the network, and lower kernel approximation error (respectively). DKS enforces these Q/C map conditions by applying a model class-preserving transformation $\hat{\phi}(x) = \gamma(\phi(\alpha x + \beta) + \delta)$. with non-trainable parameters $\alpha, \beta, \gamma$ and $\delta$. The hyperparameter $\zeta$ is chosen to be sufficiently greater than 1 (e.g. 1.5) in order to prevent the transformed activation functions from looking *"nearly linear"* (as they would be exactly linear if $\zeta = 1$), which Martens et al. (2021) argue makes it hard for the network to achieve nonlinear behavior during training. Using DKS, they were able to match the training speed of ResNets on ImageNet with vanilla networks using K-FAC. However, DKS is not fully compatible with ReLUs, and the networks in their experiments fell substantially short of ResNets in terms of generalization performance.

## 4 TAILORED ACTIVATION TRANSFORMATION (TAT)

The reason why DKS is not fully compatible with ReLUs is that they are positive homogeneous, i.e. $\phi(\alpha x) = \alpha \phi(x)$ for $\alpha \geq 0$. This makes the $\gamma$ parameter of the transformed activation function redundant, thus reducing the degrees of freedom with which to enforce DKS's four Q/C map conditions. Martens et al. (2021) attempt to circumvent this issue by dropping the condition $\mathcal{Q}'(1) = 1$, which leads to vanilla deep networks that are trainable, but slower to optimize compared to using DKS with other activation functions. This is a significant drawback for DKS, as the best generalizing deep models often use ReLU-family activations. We therefore set out to investigate other possible remedies – either in the form of different activation functions, new Q/C map conditions, or both. To this end, we adopt a ReLU-family activation function with an extra degree of freedom (known as "Leaky ReLU"), and modify the Q/C map conditions in order to preserve certain desirable properties

**Table 1:** Comparison of different methods applied to a network $f$.

| EOC (smooth) | EOC (LReLU) | DKS | TAT (smooth) | TAT (LReLU) |
|---|---|---|---|---|
| | | $\mathcal{Q}(1) = 1$ | $\mathcal{Q}(1) = 1$ | |
| $q^\infty$ exists | $\mathcal{Q}(q) = q$ | $\mathcal{Q}'(1) = 1$ | $\mathcal{Q}'(1) = 1$ | $\mathcal{Q}(q) = q$ |
| $\mathcal{C}'(1, q^\infty, q^\infty) = 1$ | $\mathcal{C}'(1) = 1$ | $\mathcal{C}_f(0) = 0$ | $\mathcal{C}'_f(1) = 1$ | $\mathcal{C}'_f(1) = 1$ |
| | | $\mathcal{C}'_f(1) = \zeta$ | $\mathcal{C}''_f(1) = \tau$ | $\mathcal{C}_f(0) = \eta$ |

of this choice. The resulting method, which we name Tailored Activation Transformation (TAT) achieves competitive generalization performance with ResNets in our experiments.

## 4.1 Tailored Activation Transformation for Leaky ReLUs

One way of addressing the issue of DKS's partial incompatibility with ReLUs is to consider a slightly different activation function – namely the Leaky ReLU (LReLU) (Maas et al., 2013):

$$\phi_\alpha(x) = \max\{x, 0\} + \alpha \min\{x, 0\}, \tag{8}$$

where $\alpha$ is the *negative slope parameter*. While using LReLUs with $\alpha \neq 0$ in place of ReLUs changes the model class, it doesn't limit the model's expressive capabilities compared to ReLU, as assuming $\alpha \neq \pm 1$, one can simulate a ReLU network with a LReLU network of the same depth by doubling the number of neurons (see Proposition 4). Rather than using a fixed value for $\alpha$, we will use it as an extra parameter to satisfy our desired Q/C map conditions. Define $\tilde{\phi}_\alpha(x) = \sqrt{\frac{2}{1+\alpha^2}}\phi_\alpha(x)$. By Lemma 1, the local Q and C maps for this choice of activation function are:

$$\mathcal{Q}(q) = q \quad \text{and} \quad \mathcal{C}(c) = c + \frac{(1-\alpha)^2}{\pi(1+\alpha^2)}\left(\sqrt{1-c^2} - c\cos^{-1}(c)\right). \tag{9}$$

Note that the condition $\mathcal{Q}(q) = q$ is actually *stronger* than DKS's Q map conditions ($\mathcal{Q}(1) = 1$ and $\mathcal{Q}'(1) = 1$), and has the potential to reduce kernel approximation errors in finite width networks compared to DKS, as it provides a better guarantee on the stability of $\mathcal{Q}_f$ w.r.t. random perturbations of the q values at each layer. Additionally, because the form of $\mathcal{C}$ does not depend on either of the layer's input q values, it won't be affected by such perturbations at all. (Notably, if one uses the negative slope parameter to transform LReLUs with DKS, these properties will *not* be achieved.) In support of these intuitions is the fact that better bounds on the kernel approximation error exist for ReLU networks than for general smooth ones (as discussed in Appendix A.1).

Another consequence of using $\tilde{\phi}_\alpha(x)$ for our activation function is that we have $\mathcal{C}'(1) = 1$ as in EOC. If combined with the condition $\mathcal{C}(0) = 0$ (which is used to achieve $\mathcal{C}_f(0) = 0$ in DKS) this would imply by Theorem 1 that $\mathcal{C}$ is the identity function, which by equation 9 is only true when $\alpha = 1$, thus resulting in a linear network. In order to avoid this situation, and the closely related one where $\tilde{\phi}_\alpha$ appears "nearly linear", we instead choose the value of $\alpha$ so that $\mathcal{C}_f(0) = \eta$, for a hyperparameter $0 \leq \eta \leq 1$. As shown in the following theorem, $\eta$ controls how close $\mathcal{C}_f$ is to the identity, thus allowing us to achieve a well-behaved global C map without making $\tilde{\phi}_\alpha$ nearly linear:

**Theorem 1.** *For a network $f$ with $\tilde{\phi}_\alpha(x)$ as its activation function (with $\alpha \geq 0$), we have*

$$\max_{c \in [-1,1]} |\mathcal{C}_f(c) - c| \leq \min\{4\mathcal{C}_f(0), 1 + \mathcal{C}_f(0)\}, \quad \max_{c \in [-1,1]} |\mathcal{C}'_f(c) - 1| \leq \min\{4\mathcal{C}_f(0), 1\} \tag{10}$$

Another motivation for using $\tilde{\phi}_\alpha(x)$ as an activation function is given by the following proposition:

**Proposition 1.** *The global C map of a feedforward network with $\tilde{\phi}_\alpha(x)$ as its activation function is equal to that of a rescaled ResNet of the same depth (see Section 2.4) with normalized ReLU activation $\phi(x) = \sqrt{2}\max(x, 0)$, shortcut weight $\sqrt{\frac{\alpha}{1+\alpha^2}}$, and residual branch $\mathcal{R}$ consisting of a combined layer (or just a normalized ReLU activation) followed by an affine layer.*

This result implies that at initialization, a vanilla network using $\tilde{\phi}_\alpha$ behaves similarly to a ResNet, a property that is quite desirable given the success that ResNets have already demonstrated.

In summary, we have the following three conditions:

$$\mathcal{Q}(q) = q, \quad \mathcal{C}'_f(1) = 1, \quad \mathcal{C}_f(0) = \eta, \tag{11}$$

which we achieve by picking the negative slope parameter $\alpha$ so that $C_f(0) = \eta$. We define the Tailored Rectifier (TReLU) to be $\tilde{\phi}_\alpha$ with $\alpha$ chosen in this way. Note that the first two conditions are

also true when applying the EOC method to LReLUs, and its only the third which sets `TReLU` apart. While this might seem like a minor difference, it actually matters a lot to the behavior of the global C map. This can be seen in Figure 2 where the c value quickly converges towards 1 with depth under EOC, resulting in a degenerate global C map. By contrast, the global C map of `TReLU` for a fixed $\eta$ converges rapidly to a nice function, suggesting a very deep vanilla network with `TReLU` has the same well-behaved global C map as a shallow network. We prove this in Proposition 3 by showing the local C map in equation 9 converges to an ODE as we increase the depth. For direct comparison of all Q/C map conditions, we refer the readers to Table 1.

For the hyperparameter $0 \le \eta \le 1$, we note that a value very close to 0 will produce a network that is "nearly linear", while a value very close to 1 will give rise to a degenerate C map. In practice we use $\eta = 0.9$ or $0.95$, which seems to work well in most settings. Once we decide on $\eta$, we can solve the value $\alpha$ using binary search by exploiting the closed-form form of $\mathcal{C}$ in equation 9 to efficiently compute $\mathcal{C}_f(0)$. For instance, if $f$ is a 100 layer vanilla network, one can compute $\mathcal{C}_f(0)$ as follows:

$$\mathcal{C}_f(0) = \overbrace{\mathcal{C} \circ \mathcal{C} \cdots \mathcal{C} \circ \mathcal{C}}^{100 \text{ times}}(0), \tag{12}$$

which is a function of $\alpha$. This approach can be generalized to more advanced architectures, such as rescaled ResNets, as discussed in Appendix B.

## 4.2 TAILORED ACTIVATION TRANSFORMATION FOR SMOOTH ACTIVATION FUNCTIONS

Unlike LReLU, most activation functions don't have closed-form formulas for their local C maps. As a result, the computation of $\mathcal{C}_f(0)$ involves the numerical approximation of many two-dimensional integrals to high precision (as in equation 5), which can be quite expensive. One alternative way to control how close $\mathcal{C}_f$ is to the identity, while maintaining the condition $\mathcal{C}'_f(1) = 1$, is to modulate its second derivative $\mathcal{C}''_f(1)$. The validity of this approach is established by the following theorem:

**Theorem 2.** *Suppose $f$ is a network with a smooth activation function. If $\mathcal{C}'_f(1) = 1$, then we have*

$$\max_{c \in [-1,1]} |\mathcal{C}_f(c) - c| \le 2\mathcal{C}''_f(1), \quad \max_{c \in [-1,1]} |\mathcal{C}'_f(c) - 1| \le 2\mathcal{C}''_f(1) \tag{13}$$

Given $\mathcal{C}(1) = 1$ and $\mathcal{C}'(1) = 1$, a straightforward computation shows that $\mathcal{C}''_f(1) = L\mathcal{C}''(1)$ if $f$ is an $L$-layer vanilla network. (See Appendix B for a discussion of how to do this computation for more general architectures.) From this we obtain the following four local Q/C map conditions:

$$\mathcal{Q}(1) = 1, \quad \mathcal{Q}'(1) = 1, \quad \mathcal{C}''(1) = \tau/L, \quad \mathcal{C}'(1) = 1. \tag{14}$$

To achieve these we adopt the same activation transformation as DKS: $\hat{\phi}(x) = \gamma(\phi(\alpha x + \beta) + \delta)$ for non-trainable scalars $\alpha$, $\beta$, $\delta$, and $\gamma$. We emphasize that these conditions cannot be used with LReLU, as LReLU networks have $\mathcal{C}''(1) = \infty$. By equation 4 and basic properties of expectations, we have

$$1 = \mathcal{Q}(1) = \mathbb{E}_{z \sim \mathcal{N}(0,1)} \left[ \hat{\phi}(z)^2 \right] = \gamma^2 \mathbb{E}_{z \sim \mathcal{N}(0,1)} \left[ (\phi(\alpha z + \beta) + \delta)^2 \right] \tag{15}$$

so that $\gamma = \mathbb{E}_{z \sim \mathcal{N}(0,1)} \left[ (\phi(\alpha z + \beta) + \delta)^2 \right]^{-1/2}$. To obtain the values for $\alpha$, $\beta$ and $\delta$, we can treat the remaining conditions as a three-dimensional nonlinear system, which can be written as follows:

$$\begin{aligned} \mathbb{E}_{z \sim \mathcal{N}(0,1)} \left[ \hat{\phi}(z)\hat{\phi}'(z)z \right] &= \mathcal{Q}'(1) = 1, \\ \mathbb{E}_{z \sim \mathcal{N}(0,1)} \left[ \hat{\phi}''(z)^2 \right] &= \mathcal{C}''(1) = \tau/L, \ \mathbb{E}_{z \sim \mathcal{N}(0,1)} \left[ \hat{\phi}'(z)^2 \right] = \mathcal{C}'(1) = 1. \end{aligned} \tag{16}$$

We do not have a closed-form solution of this system. However, each expectation is a one dimensional integral, and so can be quickly evaluated to high precision using Gaussian quadrature. One can then use black-box nonlinear equation solvers, such as modified Powell's method (Powell, 1964), to obtain a solution. See https://github.com/deepmind/dks for a complete implementation.

## 5 EXPERIMENTS

Our main experimental evaluation of TAT and competing approaches is on training deep convolutional networks for ImageNet classification (Deng et al., 2009). The goal of these experiments is *not* to achieve state-of-the-art, but rather to compare TAT as fairly as possible with existing methods, and standard ResNets in particular. To this end, we use ResNet V2 (He et al., 2016b) as the main reference

architecture, from which we obtain rescaled ResNets (by removing normalization layers and weighing the branches as per equation 6), and vanilla networks (by further removing shortcuts). For networks without batch normalization, we add dropout to the penultimate layer for regularization, as was done in Brock et al. (2021b). We train the models with 90 epochs and a batch size of 1024, unless stated otherwise. For TReLU, we obtain $\eta$ by grid search in $\{0.9, 0.95\}$. The weight initialization used for all methods is the Orthogonal Delta initialization, with an extra multiplier given by $\sigma_w$. We initialize biases iid from $\mathcal{N}(0, \sigma_b^2)$. We use $(\sigma_w, \sigma_b) = (1, 0)$ in all experiments (unless explicitly stated otherwise), with the single exception that we use $(\sigma_w, \sigma_b) = (\sqrt{2}, 0)$ in standard ResNets, as per standard practice (He et al., 2015). For all other details see Appendix D.

## 5.1 TOWARDS REMOVING BATCH NORMALIZATION

Two crucial components for the successful training of very deep neural networks are shortcut connections and batch normalization (BN) layers. As argued in De & Smith (2020) and Shao et al. (2020), BN implicitly biases the residual blocks toward the identity function, which makes the network better behaved at initialization time, and thus easier to train. This suggests that one can compensate for the removal of BN layers, at least in terms of their effect on the behaviour of the network at initialization time, by down-scaling the residual branch of each residual block. Arguably, almost all recent work on training deep networks without normalization layers (Zhang et al., 2018; Shao et al., 2020; Bachlechner et al., 2020; Brock et al., 2021a;b) has adopted this idea by introducing multipliers on the residual branches (which may or may not be optimized during training).

In Table 2, we show that one can close most of the gap with standard ResNets by simply adopting the modification in equation 6 without using BN layers. By further replacing ReLU with TReLU, we can exactly match the performance of standard ResNets. With K-FAC as the optimizer, the rescaled ResNet with shortcut weight $w = 0.9$

**Table 2:** Top-1 validation accuracy of rescaled ResNet50 with varying shortcut weights. We set $\eta = 0.9$ for TReLU.

| Optimizer | Standard ResNet | Activation | Rescaled ResNet ($w$) | | | |
|---|---|---|---|---|---|---|
| | | | 0.0 | 0.5 | 0.8 | 0.9 |
| K-FAC | 76.4 | ReLU | 72.6 | 74.5 | 75.6 | 75.9 |
| | | TReLU | 74.6 | 75.5 | **76.4** | 75.9 |
| SGD | 76.3 | ReLU | 63.7 | 72.4 | 73.9 | 75.0 |
| | | TReLU | 71.0 | 72.6 | **76.0** | 74.8 |

is only 0.5 shy of the validation accuracy (76.4) of the standard ResNet. Further replacing ReLU with TReLU, we match the performance of standard ResNet with shortcut weight $w = 0.8$.

## 5.2 THE DIFFICULTY OF REMOVING SHORTCUT CONNECTIONS

While the aforementioned works have shown that it is possible to achieve competitive results without normalization layers, they all rely on the use of shortcut connections to make the network look more linear at initialization. A natural question to ask is whether normalization layers could compensate for the removal of shortcut connections. We address this question by training shortcut-free networks with either

**Table 3:** ImageNet top-1 validation accuracies of shortcut-free networks on ImageNet.

| Depth | Optimizers | vanilla | BN | LN |
|---|---|---|---|---|
| 50 | K-FAC | 72.6 | 72.8 | 72.7 |
| | SGD | 63.7 | 72.6 | 58.1 |
| 101 | K-FAC | 71.8 | 67.6 | 72.0 |
| | SGD | 41.6 | 43.4 | 28.6 |

BN or Layer Normalization (LN) layers. As shown in Table 3, these changes do not seem to make a significant difference, especially with strong optimizers like K-FAC. These findings are in agreement with the analyses of Yang et al. (2019) and Martens et al. (2021), who respectively showed that deep shortcut-free networks with BN layers still suffer from exploding gradients, and deep shortcut-free networks with LN layers still have degenerate C maps.

## 5.3 TRAINING DEEP NEURAL NETWORKS WITHOUT SHORTCUTS

The main motivation for developing TAT is to help deep vanilla networks achieve generalization performance similar to standard ResNets. In our investigations we include rescaled ResNets with a shortcut weight of either 0 (i.e. vanilla networks) or 0.8. In Table 4 we can see that with a strong optimizer like K-FAC, we can reduce the gap on the 50 layer network to only 1.8% accuracy when training for 90 epochs, and further down to 0.6% when training for 180 epochs. For 101 layers, the gaps are 3.6% and 1.7% respectively, which we show can be further reduced with wider networks (see Table 9). To our knowledge, this is the first time that a deep vanilla network has been trained to such a high validation accuracy on ImageNet. In addition, our networks have fewer parameters and run faster than standard ResNets, and use less memory at inference time due to the removal of

**Table 4:** ImageNet top-1 validation accuracy. For rescaled ResNets ($w = 0.0$ or $w = 0.8$), we do not include any normalization layer. For standard ResNets, batch normalization is included. By default, ReLU activation is used for standard ResNet while we use `TReLU` for rescaled networks.

| Depth | Optimizer | 90 epochs | | | 180 epochs | | |
|---|---|---|---|---|---|---|---|
| | | ResNet | $w = 0.0$ | $w = 0.8$ | ResNet | $w = 0.0$ | $w = 0.8$ |
| 50 | K-FAC | 76.4 | 74.6 | 76.4 | 76.6 | 76.0 | 77.0 |
| | SGD | 76.3 | 71.0 | 76.0 | 76.6 | 72.3 | 76.8 |
| 101 | K-FAC | 77.8 | 74.2 | 77.8 | 77.6 | 75.9 | 78.4 |
| | SGD | 77.9 | 70.0 | 77.3 | 77.6 | 73.8 | 77.4 |

shortcut connections and BN layers. The gaps when using SGD as the optimizer are noticeably larger, which we further explore in Section 5.5. Lastly, using rescaled ResNets with a shortcut weight of $0.8$ and `TReLU`, we can exactly match or even surpass the performance of standard ResNets.

## 5.4 COMPARISONS WITH EXISTING APPROACHES

**Comparison with EOC.** Our first comparison is between TAT and EOC on vanilla deep networks. For EOC with ReLUs we set $(\sigma_w, \sigma_b) = (\sqrt{2}, 0)$ to achieve $\mathcal{Q}(1) = 1$ as in He et al. (2015), since ReLU networks always satisfy $\mathcal{C}'(1) = 1$ whenever $\sigma_b = 0$. For Tanh activations, a comprehensive comparison with EOC is more difficult, as there are infinitely many choices of $(\sigma_w, \sigma_b)$ that achieve $\mathcal{C}'(1) = 1$. Here we use $(\sigma_w, \sigma_b) = (1.302, 0.02)^2$, as suggested in Hayou et al. (2019). In Table 5, we can see that in all the settings, networks constructed

**Table 5:** ImageNet top-1 validation accuracy comparison between EOC and TAT on deep vanilla networks.

| Depth | Optimizer | Method | (L)ReLU | Tanh |
|---|---|---|---|---|
| 50 | K-FAC | EOC | 72.6 | 70.6 |
| | | TAT | **74.6** | **73.1** |
| | SGD | EOC | 63.7 | 55.7 |
| | | TAT | **71.0** | **69.5** |
| 101 | K-FAC | EOC | 71.8 | 69.2 |
| | | TAT | **74.2** | **72.8** |
| | SGD | EOC | 41.6 | 54.0 |
| | | TAT | **70.0** | **69.0** |

with TAT outperform EOC-initialized networks by a significant margin, especially when using SGD. Another observation is that the accuracy of EOC-initialized networks drops as depth increases.

**Comparison with DKS.** The closest approach to TAT in the existing literature is DKS, whose similarity and drawbacks are discussed in Section 4. We compare TAT to DKS on both LReLUs[3], and smooth functions like the SoftPlus and Tanh. For smooth activations, we perform a grid search over $\{0.2, 0.3, 0.5\}$ for $\tau$ in TAT, and $\{1.5, 10.0, 100.0\}$ for $\zeta$ in DKS, and report only the best performing one. From the results shown in Table 7, we observe that TAT, together with LReLU (i.e. `TReLU`), performs the best in nearly all settings we tested, and that its advantage becomes larger when we remove dropout. One possible reason for the superior performance of `TReLU` networks is the stronger Q/C map conditions that they satisfy compared to other activations (i.e. $\mathcal{Q}(q) = q$ for all $q$ vs $\mathcal{Q}(1) = 1$ and $\mathcal{Q}'(1) = 1$, and invariance of $\mathcal{C}$ to the input q value), and the extra resilience to kernel approximation error that these stronger conditions imply. In practice, we found that `TReLU` indeed has smaller kernel approximation error (compared to DKS with smooth activation functions, see Appendix E.1) and works equally well with Gaussian initialization (see Appendix E.7).

**Comparison with PReLU.** The Parametric ReLU (PReLU) introduced in He et al. (2015) differs from LReLU by making the negative slope a trainable parameter. Note that this is distinct from what we are doing with `TReLU`, since there we compute the negative slope parameter ahead of time and fix it during training. In our comparisons with PReLU we consider

**Table 6:** Comparison with PReLU.

| Depth | Optimizer | TReLU | PReLU$_{0.0}$ | PReLU$_{0.25}$ |
|---|---|---|---|---|
| 50 | K-FAC | 74.6 | 72.5 | 73.6 |
| | SGD | 71.0 | 66.7 | 67.9 |
| 101 | K-FAC | 74.2 | 71.9 | 72.8 |
| | SGD | 70.0 | 54.3 | 66.3 |

two different initializations: $0$ (which recovers the standard ReLU), and $0.25$, which was used in He et al. (2015). We report the results on deep vanilla networks in Table 6 (see Appendix E.6 for results on rescaled ResNets). For all settings, our method outperforms PReLU by a large margin, emphasizing the importance of the initial negative slope value. In principle, these two methods can be combined together (i.e. we could first initialize the negative slope parameter with TAT, and then optimize it during training), however we did not see any benefit from doing this in our experiments.

---

[2]We also ran experiments with $(\sigma_w, \sigma_b) = (1.0, 0.0)$, and the scheme described in Pennington et al. (2017) and Xiao et al. (2018) for dynamical isometry. The results were worse than those reported in the table.

[3]For DKS, we set the negative slope as a parameter and adopt the transformation $\hat{\phi}(x) = \gamma(\phi_\alpha(x + \beta) + \delta)$.

**Table 7:** Comparisons between TAT and DKS. The numbers on the right hand of / are results without dropout. The methods with ∗ are introduced in this paper.

| Depth | Optimizer | Shortcut Weight | TAT | | | DKS | | |
|---|---|---|---|---|---|---|---|---|
| | | | LReLU* | SoftPlus* | Tanh* | LReLU* | SoftPlus | Tanh |
| 50 | K-FAC | $w = 0.0$ | **74.6/74.2** | 74.4/**74.2** | 73.1/72.9 | 74.3/74.3 | 74.3/73.7 | 72.9/72.9 |
| | | $w = 0.8$ | **76.4**/75.9 | **76.4**/75.0 | 74.8/74.4 | 76.2/**76.2** | 76.3/75.1 | 74.7/74.5 |
| | SGD | $w = 0.0$ | 71.1/71.1 | 70.2/70.0 | 69.5/69.5 | 70.4/70.4 | **71.8/71.4** | 69.2/69.2 |
| | | $w = 0.8$ | **76.0/75.8** | 74.3/73.8 | 72.4/72.2 | 73.4/73.0 | 75.2/74.1 | 72.8/72.8 |
| 101 | K-FAC | $w = 0.0$ | **74.2/74.2** | 74.1/73.4 | 72.8/72.5 | 73.5/73.5 | 73.9/73.1 | 72.5/72.4 |
| | | $w = 0.8$ | **77.8/77.0** | 76.6/75.7 | 75.8/75.1 | 76.8/76.7 | 76.8/75.6 | 75.9/75.7 |
| | SGD | $w = 0.0$ | 70.0/**70.0** | **70.3**/68.8 | 69.0/67.8 | 68.3/68.3 | 68.3/68.3 | 69.8/69.8 |
| | | $w = 0.8$ | **77.3/76.0** | 75.3/75.3 | 73.8/73.5 | 74.9/74.6 | 76.3/75.1 | 74.6/74.6 |

## 5.5 THE ROLE OF THE OPTIMIZER

One interesting phenomenon we observed in our experiments, which echoes the findings of Martens et al. (2021), is that a strong optimizer such as K-FAC significantly outperforms SGD on vanilla deep networks in terms of training speed. One plausible explanation is that K-FAC works better than SGD in the large-batch setting, and our default batch size of 1024 is already beyond SGD's "critical batch size", at which scaling efficiency begins to drop. Indeed, it was shown by Zhang et al. (2019) that optimization algorithms that employ preconditioning, such as Adam and K-FAC, result in much larger critical batch sizes.

**Table 8:** Batch size scaling.

| Optimizer | Batch size | | | | | |
|---|---|---|---|---|---|---|
| | 128 | 256 | 512 | 1024 | 2048 | 4096 |
| K-FAC | 74.5 | 74.4 | 74.5 | 74.6 | 74.2 | 72.0 |
| SGD | 72.7 | 72.6 | 72.7 | 71.0 | 69.3 | 62.0 |
| LARS | 72.4 | 72.3 | 72.6 | 71.8 | 71.3 | 70.2 |

To investigate this further, we tried batch sizes between 128 and 4096 for training 50-layer vanilla `TReLU` networks. As shown in Table 8, K-FAC performs equally well for all different batch sizes except 4096 (where we see increased overfitting), while the performance of SGD starts to drop when we increase the batch size past 512. Surprisingly, we observe a similar trend for the LARS optimizer (You et al., 2019), which was designed for large-batch training. Even at the smallest batch size we tested (128), K-FAC still outperforms SGD by a gap of 1.8% within our standard epoch budget. We conjecture the rea-

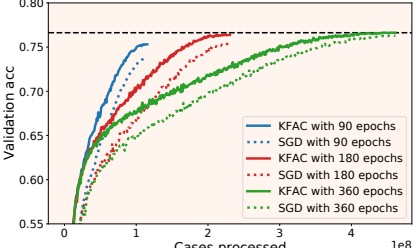

**Figure 3:** Training speed comparison between K-FAC and SGD on 50 layer vanilla `TReLU` network.

son behind this to be that vanilla networks without normalization and shortcuts give rise to loss landscapes with worse curvature properties compared to ResNets, and that this slows down simpler optimizers like SGD. To investigate further, we also ran SGD (with a batch size of 512) and K-FAC for up to 360 epochs with a "one-cycle" cosine learning rate schedule (Loshchilov & Hutter, 2016) that decreases the learning rate to to 0 by the final epoch. As shown in Figure 3, SGD does indeed eventually catch up with K-FAC (using cosine scheme), requiring just over double the number of epochs to achieve the same validation accuracy. While one may argue that K-FAC introduces additional computational overhead at each step, thus making a head-to-head comparison versus SGD unfair, we note that this overhead can amortized by not updating K-FAC's preconditioner matrix at every step. In our experiments we found that this strategy allowed K-FAC to achieve a similar per-step runtime to SGD, while retaining its optimization advantage on vanilla networks. (See Appendix E.3.)

## 6 CONCLUSIONS

In this work we considered the problem of training and generalization in vanilla deep neural networks (i.e. those without shortcut connections and normalization layers). To address this we developed a novel method that modifies the activation functions in a way tailored to the specific architecture, and which enables us to achieve generalization performance on par with standard ResNets of the same width/depth. Unlike the most closely related approach (DKS), our method is fully compatible with ReLU-family activation functions, and in fact achieves its best performance with them. By obviating the need for shortcut connections, we believe our method could enable further research into deep models and their representations. In addition, our method may enable new architectures to be trained for which existing techniques, such as shortcuts and normalization layers, are insufficient.

## REPRODUCIBILITY STATEMENT

Here we discuss our efforts to facilitate the reproducibility of this paper. Firstly, we have made an open Python implementation of DKS and TAT, supporting multiple tensor programming frameworks, available at https://github.com/deepmind/dks. Secondly, we have given all important details of our experiments in Appendix D.

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

# A  BACKGROUND

## A.1  KERNEL FUNCTION APPROXIMATION ERROR BOUNDS

In Section 2.1, we claimed that the kernel defined in equation 2 would converge to a deterministic kernel, as the width of each layer goes to infinity. To be specific, one has the following result bounding the kernel approximation error.

**Theorem 3** (Adapted from Theorem 2 of Daniely et al. (2016)). *Consider a fully-connected network of depth $L$ with weights initialized independently using a standard Gaussian fan-in initialization. Further suppose that the activation function $\phi$ is $C$-bounded (i.e. $\|\phi\|_\infty \leq C$, $\|\phi'\|_\infty \leq C$ and $\|\phi''\|_\infty \leq C$ for some constant $C$) and satisfies $\mathbb{E}_{z\sim\mathcal{N}(0,1)}[\phi(z)^2] = 1$, and that the width of each layer is greater than or equal to $(4C^4)^L \log(8L/\delta)/\epsilon^2$. Then at initialization time, for inputs $x_1$ and $x_2$ satisfying $\|x_1\|^2 = \|x_2\|^2 = \dim(x_1)$, we have that*

$$\left| \kappa_f^L(x_1, x_2) - \tilde{\kappa}_f^L(x_1, x_2) \right| \leq \epsilon$$

*with probability at least $1 - \delta$.*

The bound in Theorem 3 predicts an exponential dependence on the depth $L$ of the minimum required width of each layer. However, for a network with ReLU activations, this dependence is only quadratic in $L$, as is established in the following theorem:

**Theorem 4** (Adapted from Theorem 3 of Daniely et al. (2016)). *Consider a fully-connected network of depth $L$ with ReLU activations and weights initialized independently using a He initialization (He et al., 2015), and suppose that the width of each layer is greater than or equal to $L^2 \log(L/\delta)/\epsilon^2$. Then at initialization time, for inputs $x_1$ and $x_2$ satisfying $\|x_1\|^2 = \|x_2\|^2 = \dim(x_1)$, and $\epsilon \lesssim \frac{1}{L}$, we have that*

$$\left| \kappa_f^L(x_1, x_2) - \tilde{\kappa}_f^L(x_1, x_2) \right| \leq \epsilon$$

*with probability at least $1 - \delta$.*

According to Lemma D.1 of Buchanan et al. (2020), the requirement of the width for ReLU networks could further be reduced to linear in the depth $L$, but with a worse dependency on $\delta$.

Although Theorems 3 and 4 are only applicable to Gaussian initializations, a similar bound has been given by Martens (2021) for scaled uniform orthogonal initializations in the case that $L = 1$. Moreover, Martens (2021) conjectures that their result could be extended to general values of $L$.

## A.2  DEGENERATE C MAPS FOR VERY DEEP NETWORKS

Daniely et al. (2016), Poole et al. (2016), and Martens et al. (2021) have shown that without very careful interventions, C maps inevitably become "degenerate" in deep networks, tending rapidly towards constant functions on $(-1, 1)$ as depth increases. The following proposition is a restatement of Claim 1 from Daniely et al. (2016):

**Proposition 2.** *Suppose $f$ is a deep network consisting of a composition of $L$ combined layers. Then for all $c \in (-1, 1)$ we have*

$$\lim_{L\to\infty} \mathcal{C}_f(c) = c^*,$$

*for some $c^* \in [0, 1]$.*

While the above result doesn't characterize the *rate* of convergence $\mathcal{C}_f(c)$ to a constant function, Poole et al. (2016) show that if $\mathcal{C}'(1) \neq 1$, it happens exponentially fast as a function of $L$ in the asymptotic limit of large $L$. Martens et al. (2021) gives a similar result which holds uniformly for all $L$, and for networks with more general repeated structures.

## A.3  C MAP DERIVATIVE

Poole et al. (2016) gave the following nice formula for the derivative of C map of a combined layer with activation function $\phi$:

$$\mathcal{C}'(c, q_1, q_2) = \frac{\sqrt{q_1 q_2}}{\sqrt{\mathcal{Q}(q_1)\mathcal{Q}(q_2)}} \mathbb{E}_{z_1, z_2 \sim \mathcal{N}(0,1)} \left[ \phi'\left(\sqrt{q_1} z_1\right) \phi'\left(\sqrt{q_2}\left(c z_1 + \sqrt{1 - c^2} z_2\right)\right) \right]. \quad (17)$$

For a rigorous proof of this result we refer the reader to Martens et al. (2021).

One can iterate this formula to obtain a similar equation for higher-order derivatives:

$$\mathcal{C}^{(i)}(c, q_1, q_2) = \frac{(q_1 q_2)^{(i/2)}}{\sqrt{\mathcal{Q}(q_1)\mathcal{Q}(q_2)}} \mathbb{E}_{z_1, z_2 \sim \mathcal{N}(0,1)} \left[ \phi^{(i)}\left(\sqrt{q_1} z_1\right) \phi^{(i)}\left(\sqrt{q_2}\left(c z_1 + \sqrt{1 - c^2} z_2\right)\right)\right]. \tag{18}$$

### A.4 SOME USEFUL PROPERTIES OF C MAPS

In this section we will assume that $q_1 = q_2 = 1$.

Observe that $\mathcal{C}(1) = \mathbb{E}_{z \sim \mathcal{N}(0,1)}\left[\phi(z)^2\right] = 1$ and that $\mathcal{C}$ maps $[-1, 1]$ to $[-1, 1]$ (which follows from its interpretation as computing cosine similarities for infinitely wide networks). Moreover, $\mathcal{C}$ is a positive definite function, which means that it can be written as $\sum_{n=0}^{\infty} b_n c^n$ for $b_n \geq 0$ (Daniely et al., 2016; Martens et al., 2021). Note that for smooth activation functions, positive definiteness can be easily verified by Taylor-expanding $\mathcal{C}(c)$ about $c = 0$ and using

$$\mathcal{C}^{(i)}(0) = \mathbb{E}_{z \sim \mathcal{N}(0,1)}\left[\phi^{(i)}(z)\right]^2 \geq 0. \tag{19}$$

As discussed in Section 2.3, global C maps are computed by recursively taking compositions and weighted averages (with non-negative weights), starting from $\mathcal{C}$. Because all of the above properties are preserved under these operations, it follows that global C maps inherit them from $\mathcal{C}$.

## B ADDITIONAL DETAILS AND PSEUDOCODE FOR ACTIVATION FUNCTION TRANSFORMATIONS

### B.1 TAKING ALL SUBNETWORKS INTO ACCOUNT

In the main text of this paper we have used the condition $\mathcal{C}'_f(1) = \zeta$ in DKS, $\mathcal{C}_f(0) = \eta$ in TAT for Leaky ReLUs, and $\mathcal{C}''_f(1) = \tau$ in TAT for smooth activation functions. However, the condition used by Martens et al. (2021) in DKS was actually $\mu_f^1(\mathcal{C}'(1)) = \zeta$, where $\mu_f^1$ is the so-called "maximal slope function":

$$\mu_f^1(\mathcal{C}'(1)) = \max_{g: g \subseteq f} \mathcal{C}'_g(1),$$

where "$g \subseteq f$" denotes that $g$ is a subnetwork[4] of $f$. (That $\mathcal{C}'_g(1)$ is fully determined by $\mathcal{C}'(1)$ follows from the fact that $\mathcal{C}_g$ can be written in terms of compositions, weighted average operations, and applications of $\mathcal{C}$, and that C maps always preserve the value 1. Using the chain rule, and the linearity of derivatives, these facts allow one to write $\mathcal{C}'_g(1)$ as a polynomial function of $\mathcal{C}'(1)$.)

The motivation given by Martens et al. (2021) for looking at $\mathcal{C}'_g(1)$ over all subnetworks $g \subseteq f$ (instead of just $\mathcal{C}'_f(1)$) is that we want *all* layers of $f$, in all of its subnetworks, to be readily trainable. For example, a very deep and untrainable MLP could be made to have a reasonable global C map simply by adding a skip connection from its input to its output, but this won't do anything to address the untrainability of the layers being "skipped around" (which form a subnetwork).

In the main text we ignored this complication in the interest of a shorter presentation, and because we happened to have $\mu_f^1(\mathcal{C}'(1)) = \mathcal{C}'_f(1)$ for the simple network architectures focused on in this work. To remedy this, in the current section we will discuss how to modify the conditions $\mathcal{C}_f(0) = \eta$ and $\mathcal{C}''_f(1) = \tau$ used in TAT so that they take into account all subnetworks. This will be done using a natural generalization of the maximal slope function from DKS. We will then address the computational challenges that result from doing this.

---

[4] A *subnetwork* of $f$ is defined as a (non-strict) connected subset of the layers in $f$ that constitute a neural network with a singular input and output layer. So for example, layers 3, 4 and 5 of a 10 layer MLP form a subnetwork, while layers 3, 4, and 6 do not.

To begin, we will replace the condition $\mathcal{C}_f(0) = \eta$ (used in TAT for Leaky ReLUs) by the condition $\mu_f^0(0) = \eta$, where we define the *maximal c value function* $\mu_f^0$ of $f$ by

$$\mu_f^0(\alpha) = \max_{g:g \subseteq f} \mathcal{C}_g(0),$$

where $\alpha$ is the negative slope parameter (which determines $\mathcal{C}$ in LReLU networks [via $\phi_\alpha$] and thus each $\mathcal{C}_g$).

We will similarly replace the condition $\mathcal{C}_f''(1) = \tau$ (used in TAT for smooth activations) by the condition $\mu_f^2(\mathcal{C}''(1)) = \tau$, where we define the *maximal curvature function* $\mu_f^2$ of $f$ by

$$\mu_f^2(\mathcal{C}''(1)) = \max_{g:g \subseteq f} \mathcal{C}_g''(1),$$

where each $\mathcal{C}_g''(1)$ is determined by $\mathcal{C}''(1)$. That each $\mathcal{C}_g''(1)$ is a well-defined function of $\mathcal{C}''(1)$ follows from the fact that C maps always map the value 1 to 1, the aforementioned relationship between $\mathcal{C}_g$ and $\mathcal{C}$, and the fact that we have $\mathcal{C}'(1) = 1$ under TAT (so that $\mathcal{C}_h'(1) = 1$ for all subnetworks $h$). These facts allow us to write $\mathcal{C}_g''(1)$ as a constant multiple of $\mathcal{C}''(1)$ using the linearity of 2nd derivatives and the 2nd-order chain rule (which is given by $(a \circ b)''(x) = a''(b(x))b'(x)^2 + a'(b(x))b''(x)$).

### B.2    COMPUTING $\mu_f^0$ AND $\mu_f^2$ IN GENERAL

Given these new conditions for TAT, it remains to compute their left hand sides so that we may ultimately solve for the required quantities ($\alpha$ or $\mathcal{C}''(1)$). In Section 2.3 we discussed how a (sub)network $f$'s C map $\mathcal{C}_f$ can be computed in terms of the local C map $\mathcal{C}$ by a series of composition and non-negative weighted sum operations. We can define a generalized version of this construction $U_{f,r}$ which replaces $\mathcal{C}$ with an arbitrary non-decreasing function $r$, so that $U_{f,\mathcal{C}}(c) = \mathcal{C}_f(c)$. A recipe for computing $U_{f,r}$ is given in Appendix B.4.

Given $U_{f,r}$, we define the *subnetwork maximizing function* $M$ by

$$M_{f,r}(x) = \max_{g:g \subseteq f} U_{g,r}(x).$$

With this definition, it is not hard to see that if $r_0(x) = \mathcal{C}(x)$, $r_1(x) = \mathcal{C}'(1)x$, and $r_2(x) = \mathcal{C}''(1)+x$, then $\mu_f^0(\alpha) = M_{f,r_0}(0)$ (where the dependence on $\alpha$ is implicit through the dependence of $\mathcal{C}$ on $\phi_\alpha$), $\mu_f^1(\mathcal{C}'(1)) = M_{f,r_1}(1)$, and $\mu_f^2(\mathcal{C}''(1)) = M_{f,r_2}(0)$. Thus, it suffices to derive a scheme for computing (and inverting) $M_{f,r}$ for general networks $f$ and non-decreasing functions $r$.

Naively, computing $M_{f,r}$ could involve a very large maximization and be quite computationally expensive. But analogously to the maximal slope function computation described in Martens et al. (2021), the computation of $M_{f,r}$ can simplified substantially, so that we rarely have to maximize over more than a few possible subnetworks. In particular, since $U_{g,r}(x)$ is a non-decreasing function of $x$ for all $g$ (which follows from the fact that $r$ is non-decreasing), and $U_{g \circ h,r} = U_{g,r} \circ U_{h,r}$, it thus follows that $U_{g \circ h,r}(x) \geq U_{g,r}(x), U_{h,r}(x)$ for all $x$. This means that for the purposes of the maximization, we can ignore any subnetwork in $f$ which composes with another subnetwork (not necessarily in $f$) to form a strictly larger subnetwork isomorphic to one in $f$. This will typically be the vast majority of them. Note that this does *not* therefore imply that $M_{f,r} = U_{f,r}$, since not all subnetworks compose in this way. For example, a sufficiently deep residual branch of a residual block in a rescaled ResNet won't compose with *any* subnetwork to form a larger one.

### B.3    SOLVING FOR $\alpha$ AND $\mathcal{C}''(1)$

Having shown how to efficiently compute $M_{f,r}$, and thus both of $\mu_f^0$ and $\mu_f^2$, it remains to show how we can invert them to find solutions for $\alpha$ and $\mathcal{C}''(1)$ (respectively). Fortunately, this turns out to be easy, as both functions are strictly monotonic in their arguments ($\alpha$ and $\mathcal{C}''(1)$), provided that $f$ contains at least one nonlinear layer. Thus, we may apply a simple 1-dimensional root-finding approach, such as binary search.

To see that $\mu_f^0(\alpha)$ is a strictly decreasing function of $\alpha$ (or in other words, a strictly *increasing* function of $-\alpha$), we observe that it is a maximum over terms of the form $U_{g,\mathcal{C}}(0)$, which are all either strictly decreasing non-negative functions of $\alpha$, or are identically zero. These properties of $U_{g,\mathcal{C}}(0)$

follow from the fact that it involves only applications of $\mathcal{C}$, along with compositions and non-negative weighted averages, and that $\mathcal{C}(c)$ is a strictly decreasing function of $\alpha$ for all $c \in [-1, 1]$ (in Leaky ReLU networks). A similar argument can be used to show that $\mu_f^2(\mathcal{C}''(1))$ is a strictly increasing function of $\mathcal{C}''(1)$ (and is in fact equal to a non-negative multiple of $\mathcal{C}''(1)$).

## B.4   RECIPE FOR COMPUTING $U_{f,r}$

As defined, $U_{f,r}$ is computed from $f$ by taking the computational graph for $C_f$ and replacing the local C map $\mathcal{C}$ with $r$ wherever the former appears. So in particular, one can obtain a computational graph for $U_{f,r}(x)$ from $f$'s computational graph by recursively applying the following rules:

1. Composition $g \circ h$ of two subnetworks $g$ and $h$ maps to $U_{g,r} \circ U_{h,r}$.
2. Affine layers map to the identity function.
3. Nonlinear layers map to $r$.
4. Normalized sums with weights $w_1, w_2, ..., w_n$ over the outputs of subnetworks $g_1, g_2, ..., g_n$, map to the function

$$w_1^2 U_{g_1,r}(x_1) + w_2^2 U_{g_2,r}(x_2) + \cdots + w_n^2 U_{g_n,r}(x_n),$$

   where $x_1, x_2, ..., x_n$ are the respective inputs to the $U_{g_i,r}$'s.
5. $f$'s input layer maps to $x$.

In the special case of computing $U_{f,r_2}(0)$, one gets the following simplified list of rules:

1. Composition $g \circ h$ of two subnetworks $g$ and $h$ maps to $U_{g,r_2}(0) + U_{h,r_2}(0)$
2. Affine layers map to 0.
3. Nonlinear layers map to $\mathcal{C}''(1)$.
4. Normalized sums with weights $w_1, w_2, ..., w_n$ over the outputs of subnetworks $g_1, g_2, ..., g_n$, map to the function

$$w_1^2 U_{g_1,r_2}(0) + w_2^2 U_{g_2,r_2}(0) + \cdots + w_n^2 U_{g_n,r_2}(0).$$

5. $f$'s input layer maps to $x$.

Note that this second procedure will always produce a non-negative multiple of $\mathcal{C}''(1)$, provided that $f$ contains at least one nonlinear layer.

## B.5   RESCALED RESNET EXAMPLE

In this subsection we will demonstrate how to apply the above rules to compute the maximal curvature function $\mu_f^2$ for a rescaled ResNet $f$ with shortcut weight $w$ and residual branch $\mathcal{R}$ (as defined in equation 6). We note that this computation also handles the case of a vanilla network by simply taking $w = 0$.

First, we observe that all subnetworks in $f$ compose to form larger ones in $f$, except for $f$ itself, and for the residual branches of its residual blocks. We thus have that $\mu_f^2(\mathcal{C}''(1)) = \max\{U_{f,r_2}(0), U_{\mathcal{R},r_2}(0)\}$.

Because each residual branch has a simple feedforward structure with three nonlinear layers, it follows that $U_{\mathcal{R},r_2}(0) = 3\mathcal{C}''(1)$. And because each shortcut branch $\mathcal{S}$ has no nonlinear layers, it follows that $U_{\mathcal{S},r_2}(0) = 0$. Applying the rule for weighted averages to the output of each block $\mathcal{B}$ we thus have that $U_{\mathcal{B},r_2}(0) = w^2 U_{\mathcal{S},r_2}(0) + (1 - w^2)U_{\mathcal{R},r_2}(0) = 3(1 - w^2)\mathcal{C}''(1)$. Given a network with $L$ nonlinear layers, we have $L/3$ blocks, and since the blocks compose in a feedforward manner it thus follows that $U_{f,r_2}(0) = (L/3) \cdot 3(1 - w^2)\mathcal{C}''(1) = L(1 - w^2)\mathcal{C}''(1)$. We therefore conclude that $\mu_f^2(\mathcal{C}''(1)) = \max\{3, L(1 - w^2)\}\mathcal{C}''(1)$.

The rescaled ResNets used in our experiments have a slightly more complex structure (based on the ResNet-50 and ResNet-101 architectures), with a nonlinear layer appearing after the sequence of residual blocks, and with a four of their blocks being "transition blocks", whose shortcut branches

contain a nonlinear layer. In these networks, the total number of residual blocks is given by $(L-2)/3$. Following a similar argument to the one above we have that

$$U_{f,r_2}(0) = \left(\frac{L-2}{3} - 4\right) \cdot 3(1-w^2)\mathcal{C}''(1) + 4 \cdot (w^2 + 3(1-w^2))\mathcal{C}''(1) + \mathcal{C}''(1)$$
$$= [(L-2)(1-w^2) + 4w^2 + 1]\mathcal{C}''(1) = [(L-6)(1-w^2) + 5]\mathcal{C}''(1),$$

and thus

$$\mu_f^2(\mathcal{C}''(1)) = \max\{[(L-6)(1-w^2) + 5]\mathcal{C}''(1), 3(1-w^2)\mathcal{C}''(1)\}$$
$$= [(L-6)(1-w^2) + 5]\mathcal{C}''(1).$$

### B.6 PSEUDOCODE

---

**Algorithm 1** TAT for LReLU.

---

**Require:** The target value $\eta$ for $\mu_f^0(\alpha)$
1: Use the steps from Subsection B.2 to construct a procedure for computing the maximal c value function $\mu_f^0(\alpha)$ for general $\alpha \geq 0$. Note that the local C map $\mathcal{C}$, on which $\mu_f^0(\alpha)$ depends, can be computed efficiently for (transformed) LReLUs using equation 9.
2: Perform a binary search to find the negative slope $\alpha$ such that $\mu_f^0(\alpha) = \eta$.
3: Using the found $\alpha$, output the transformed activation function given by $\tilde{\phi}_\alpha(x) = \sqrt{\frac{2}{1+\alpha^2}}\phi_\alpha(x)$.

---

**Algorithm 2** TAT for smooth activations.

---

**Require:** The target value $\tau$ of $\mathcal{C}_f''(1)$
**Require:** The original activation function $\phi(x)$
1: Use the steps from Subsection B.2 to construct a procedure for computing the maximal curvature function $\mu_f^2(\mathcal{C}''(1))$ for general $\mathcal{C}''(1) \geq 0$.
2: Perform a binary search to find $\mathcal{C}''(1)$ such that $\mu_f^2(\mathcal{C}''(1)) = \tau$.
3: Using a numerical solver, solve the three-dimensional nonlinear system in equation 16 (but with the value of $\mathcal{C}''(1)$ found above instead of $\tau/L$) to obtain values for $\alpha$, $\beta$, $\gamma$, and $\delta$.
4: Using the solution from the last step, output the transformed activation function given by $\hat{\phi}(x) = \gamma(\phi(\alpha x + \beta) + \delta)$.

---

## C TECHNICAL RESULTS AND PROOFS

**Lemma 1.** *For networks using the activation function $\tilde{\phi}_\alpha(x) = \sqrt{\frac{2}{1+\alpha^2}}\phi_\alpha(x)$, the local Q and C maps are given by*

$$\mathcal{Q}(q) = q \quad and \quad \mathcal{C}(c) = c + \frac{(1-\alpha)^2}{\pi(1+\alpha^2)}\left(\sqrt{1-c^2} - c\cos^{-1}(c)\right). \tag{20}$$

*Proof.* In this proof we will use the notation $\mathcal{Q}_\phi$ and $\mathcal{C}_\phi$ to denote the local Q and C maps for networks that use a given activation function $\phi$.

First, we note that LReLU is basically the weighted sum of identity and ReLU. In particular, we have the following equation:

$$\phi_\alpha(x) = \alpha x + (1-\alpha)\phi_0(x) = \max\{x, 0\} + \alpha\min\{x, 0\}.$$

Second, we have that $\mathcal{Q}_{\phi_\alpha}(q) = \mathbb{E}_{z \sim \mathcal{N}(0,1)}\left[qz^2\mathbb{I}[z \geq 0]\right] + \alpha^2\mathbb{E}_{z \sim \mathcal{N}(0,1)}\left[qz^2\mathbb{I}[z < 0]\right] = \frac{1+\alpha^2}{2}q$ (from which $\mathcal{Q}_{\tilde{\phi}_\alpha}(q) = q$ immediately follows).

It then follows from equation 5, and the fact that local C maps are invariant to multiplication of the activation function by a constant, that

$$
\begin{aligned}
\mathcal{C}_{\tilde{\phi}_\alpha}(c) = \mathcal{C}_{\phi_\alpha}(c) &= \frac{2}{1+\alpha^2} \mathbb{E}_{z_1, z_2 \sim \mathcal{N}(0,1)} \left[ \phi_\alpha(z_1) \phi_\alpha \left( cz_1 + \sqrt{1-c^2} z_2 \right) \right] \\
&= \frac{2}{1+\alpha^2} \left[ \alpha^2 c + (1-\alpha)^2 \mathcal{C}_{\phi_0}(c) Q_{\phi_0}(1) \right] \\
&\quad + \frac{2}{1+\alpha^2} \left[ 2\alpha(1-\alpha) \mathbb{E}_{z_1, z_2 \sim \mathcal{N}(0,1)} \left[ (cz_1 + \sqrt{1-c^2} z_2) \phi_0(z_1) \right] \right]
\end{aligned}
\tag{21}
$$

From Daniely et al. (2016) we have that

$$
\mathcal{C}_{\phi_0}(c) = \frac{\sqrt{1-c^2} + (\pi - \cos^{-1}(c))c}{\pi},
\tag{22}
$$

and for the last part of equation 21 we have

$$
\mathbb{E}_{z_1, z_2 \sim \mathcal{N}(0,1)} \left[ (cz_1 + \sqrt{1-c^2} z_2) \phi_0(z_1) \right] = \mathbb{E}_{z_1 \sim \mathcal{N}(0,1)} \left[ cz_1^2 \mathbb{1}_{z_1 > 0} \right] = \frac{c}{2}.
\tag{23}
$$

Plugging equation 22 and equation 23 back into equation 21, we get

$$
\begin{aligned}
\mathcal{C}_{\tilde{\phi}_\alpha}(c) &= \frac{2}{1+\alpha^2} \left[ \alpha^2 c + \frac{(1-\alpha)^2}{2} \frac{\sqrt{1-c^2} + (\pi - \cos^{-1}(c))c}{\pi} + \alpha(1-\alpha)c \right] \\
&= \frac{(1-\alpha)^2 \left( \sqrt{1-c^2} + c(\pi - \cos^{-1}(c)) \right) + 2\pi\alpha c}{(1+\alpha^2)\pi}.
\end{aligned}
\tag{24}
$$

Rearranging this gives the claimed formula. ☐

**Proposition 1.** *The global C map of a feedforward network with $\tilde{\phi}_\alpha(x)$ as its activation function is equal to that of a rescaled ResNet of the same depth (see Section 2.4) with normalized ReLU activation $\phi(x) = \sqrt{2}\max(x,0)$, shortcut weight $\sqrt{\frac{\alpha}{1+\alpha^2}}$, and residual branch $\mathcal{R}$ consisting of a combined layer (or just a normalized ReLU activation) followed by an affine layer.*

*Proof.* By equation 7, the C map for a residual block $\mathcal{B}$ of the hypothesized rescaled ResNet is given by

$$
\mathcal{C}_{\mathcal{B}}(c) = w^2 c + (1 - w^2) \mathcal{C}_{\phi_0}(c).
\tag{25}
$$

The global C map of this network is given by $L$ compositions of this function, while the global C map of the hypothesized feedforward network is given by $L$ compositions of $\mathcal{C}_{\tilde{\phi}_\alpha}(c)$. So to prove the claim it suffices to show that $\mathcal{C}_{\mathcal{B}}(c) = \mathcal{C}_{\tilde{\phi}_\alpha}(c)$.

Taking $w = \sqrt{\frac{2\alpha}{1+\alpha^2}}$, one obtains the following

$$
\mathcal{C}_{\mathcal{B}}(c) = \frac{2\alpha}{1+\alpha^2} + \frac{(1-\alpha^2)}{1+\alpha^2} \frac{\sqrt{1-c^2} + c(\pi - \cos^{-1}(c))}{\pi},
\tag{26}
$$

which is exactly the same as $\mathcal{C}_{\tilde{\phi}_\alpha}(c)$ as given in Lemma 1. This concludes the proof. ☐

**Proposition 3.** *Suppose $f$ is vanilla network consisting of $L$ combined layers with the `TReLU` activation function (so that $\mathcal{C}_f(0) = \eta \in (0,1)$). Then $\mathcal{C}_f$ converges to a limiting map on $(-1,1)$ as $L$ goes to infinity. In particular,*

$$
\lim_{L \to \infty} \mathcal{C}_f(c) = \psi(c, T),
\tag{27}
$$

*where $T$ is such that $\psi(0, T) = \eta$, and where $\psi$ is the solution of the following ordinary differential equation (ODE) with the first argument being the initial condition (i.e. $\psi(c, 0) = c$), and the second argument being time:*

$$
\frac{dx(t)}{dt} = \sqrt{1 - x(t)^2} - x(t) \cos^{-1}(x(t)).
\tag{28}
$$

*Proof.* First, we notice that the local C map for `TReLU` networks can be written as a difference equation:

$$\mathcal{C}(c) = c + \frac{(1-\alpha)^2}{\pi(1+\alpha^2)}\left(\sqrt{1-c^2} - c\cos^{-1}(c)\right). \tag{29}$$

Importantly, $\mathcal{C}$ is a monotonically increasing function of $c$, whose derivative goes to zero only as $\alpha \in [0,1]$ goes to 1. Thus, to achieve $\mathcal{C}_f(0) = \eta$ in the limit of large $L$, we require that $\frac{(1-\alpha)^2}{\pi(1+\alpha^2)}$ goes to 0. This implies that the above difference equation converges to the ODE in equation 28.

Because the function $\sqrt{1-x^2} - x\cos^{-1}(x)$ is continuously differentiable in $[-1,1]$, and its derivative $-\cos^{-1}(x)$ is bounded, one can immediately show that it is globally Lipschitz, and the ODE has a unique solution $\psi(c_0, t)$ according to Theorem 3.2 of Khalil (2008).

Now, we are only left to find the time $T$ such that $\mathcal{C}_f^\infty(0) = \psi(0, T) = \eta$. To that end, we notice that

$$g(x) = \sqrt{1-x^2} - x\cos^{-1}(x) > 0, \text{ for } x \in (-1,1) \tag{30}$$

because $g(1) = 0$ and $g'(x) = -\cos^{-1}(x) < 0$ on $(-1,1)$. This implies that the $\psi(0,t)$ is a monotonically increasing continuous function of $t$. Since $\psi(0,0) = 0$, to establish the existence of $T$ it suffices to show that $\psi(0,\infty) \geq 1$.

To this end we first observe that

$$g(x) \geq \tfrac{2\sqrt{2}}{3}(1-x)^{3/2}, \tag{31}$$

which follows by defining $h(x) = g(x) - \frac{2\sqrt{2}}{3}(1-x)^{3/2}$ and observing that $h(1) = 0$ and $h'(x) = -\cos^{-1}(x) + \sqrt{2}(1-x)^{1/2} < 0$ on $(-1,1)$. Given this, it is sufficient to show that the solution $\tilde{\psi}$ for the ODE $\dot{x} = \frac{2\sqrt{2}}{3}(1-x)^{3/2}$ satisfies $\tilde{\psi}(0,\infty) = 1$. The solution $\tilde{\psi}$ turns out to have a closed-form of $\tilde{\psi}(0,t) = 1 - (\frac{3}{\sqrt{2}t+3})^2$, and thus $\psi(0,\infty) \geq \tilde{\psi}(0,\infty) = 1$. This completes the proof. $\qquad\square$

**Theorem 1.** *For a network $f$ with $\tilde{\phi}_\alpha(x)$ as its activation function (with $\alpha \geq 0$), we have*

$$\max_{c\in[-1,1]} |\mathcal{C}_f(c) - c| \leq \min\{4\mathcal{C}_f(0), 1 + \mathcal{C}_f(0)\}, \quad \max_{c\in[-1,1]} |\mathcal{C}'_f(c) - 1| \leq \min\{4\mathcal{C}_f(0), 1\} \tag{10}$$

*Proof.* Because $\mathcal{C}_f$ is a positive definite function (by Section A.4) we have that it can be written as $\mathcal{C}_f(c) = \sum_n^\infty b_n c^n$ for $b_n \geq 0$. Given $\mathcal{C}_f(1) = \mathcal{C}'_f(1) = 1$, we have

$$\sum_{n=0}^\infty b_n = \sum_{n=1}^\infty n b_n = 1 \implies b_0 = \sum_{n=2}^\infty (n-1)b_n \implies 2b_0 + b_1 \geq 1. \tag{32}$$

Hence, $1 - \mathcal{C}'_f(0) = 1 - b_1 \leq 2b_0 = 2\mathcal{C}_f(0)$. Now we are ready to bound the deviation of $\mathcal{C}_f(c)$ from identity:

$$\begin{aligned}
\max_{c\in[-1,1]} |\mathcal{C}_f(c) - c| &= \max_{c\in[-1,1]} \left| b_0 + \sum_{n=2}^\infty b_n c^n - (1-b_1)c \right| \\
&\leq \max_{c\in[-1,1]} \left[ b_0 + \sum_{n=2}^\infty b_n |c|^n + (1-b_1)|c| \right] \\
&= b_0 + \sum_{n=2}^\infty b_n + 1 - b_1 = 2(1-b_1) \\
&= 2(1 - \mathcal{C}'_f(0)) \leq 4\mathcal{C}_f(0).
\end{aligned} \tag{33}$$

Using equation 20 we have that

$$\mathcal{C}'(c) = 1 - \frac{(1-\alpha)^2}{(1+\alpha^2)\pi}\cos^{-1}(c).$$

From our assumption that $\alpha \geq 0$ it follows that $0 \leq \mathcal{C}'(c) \leq 1$ for all $c \in [-1,1]$. Since the property of having a derivative bounded between 0 and 1 is closed under functional composition and positive

weighted averages, it thus follows that $0 \le \mathcal{C}_f(c) \le 1$ for all $c \in [-1,1]$. An immediate consequence of this is that $\mathcal{C}_f(c)$ is non-decreasing, and that

$$\max_{c \in [-1,1]} |\mathcal{C}_f(c) - c| = \mathcal{C}_f(-1) + 1 \le \mathcal{C}_f(0) + 1. \tag{34}$$

Next, we bound the deviation of $\mathcal{C}'_f(c)$ from 1:

$$
\begin{aligned}
\max_{c \in [-1,1]} \left| \mathcal{C}'_f(c) - 1 \right| &= \max_{c \in [-1,1]} \left| \sum_{n=2}^{\infty} n b_n c^{n-1} - (1 - b_1) \right| \\
&\le \max_{c \in [-1,1]} \left[ \sum_{n=2}^{\infty} n b_n |c|^{n-1} + (1 - b_1) \right] \\
&= \sum_{n=2}^{\infty} n b_n + 1 - b_1 = 2(1 - b_1) \\
&= 2(1 - \mathcal{C}'_f(0)) \le 4\mathcal{C}_f(0).
\end{aligned}
\tag{35}
$$

From the previous fact that $0 \le \mathcal{C}'_f(c) \le 1$ for all $c \in [-1,1]$ we also have that $\max_{c \in [-1,1]} \left| \mathcal{C}'_f(c) - 1 \right| \le 1$. This completes the proof. $\qquad \square$

**Theorem 2.** *Suppose $f$ is a network with a smooth activation function. If $\mathcal{C}'_f(1) = 1$, then we have*

$$\max_{c \in [-1,1]} |\mathcal{C}_f(c) - c| \le 2\mathcal{C}''_f(1), \quad \max_{c \in [-1,1]} \left| \mathcal{C}'_f(c) - 1 \right| \le 2\mathcal{C}''_f(1) \tag{13}$$

*Proof.* $\mathcal{C}_f$ is a positive definite function by Section A.4. So by the fact that positive definite functions are non-negative, non-decreasing, and convex on the non-negative part of their domain, we obtain that $\mathcal{C}'_f(0) \ge \mathcal{C}'_f(1) - \mathcal{C}''_f(1) = 1 - \mathcal{C}''_f(1)$. By equation 33, we have

$$\max_{c \in [-1,1]} |\mathcal{C}_f(c) - c| \le 2(1 - \mathcal{C}'_f(0)) \le 2\mathcal{C}''_f(1). \tag{36}$$

Further by equation 35, we also have

$$\max_{c \in [-1,1]} \left| \mathcal{C}'_f(c) - 1 \right| \le 2(1 - \mathcal{C}'_f(0)) \le 2\mathcal{C}''_f(1). \tag{37}$$

This completes the proof. $\qquad \square$

**Proposition 4.** *Suppose $f$ is some function computed by a neural network with the ReLU activation. Then for any negative slope parameter $\alpha \ne \pm 1$, we can compute $f$ using an LReLU neural network of the same structure and double the width of the original network.*

*Proof.* The basic intuition behind this proof is that a ReLU unit can always be "simulated" by two LReLU units as long as $\alpha \ne \pm 1$, due to the following formula:

$$\phi_0(x) = \frac{1}{1 - \alpha^2} \left( \phi_\alpha(x) + \alpha \phi_\alpha(-x) \right).$$

We will begin by proving the claim in the case of a network with one hidden layer. In particular, we assume the ReLU network has $m$ hidden units:

$$f(w, b, a, x) = \sum_{r=1}^{m} a_r \phi_0(w_r^\top x + b_r), \tag{38}$$

where $x \in \mathbb{R}^d$ is the input, and $w \in \mathbb{R}^{md}$, $b \in \mathbb{R}^m$ and $a \in \mathbb{R}^m$ are weights, biases of the input layer and weights of output layer, respectively. For LReLU with negative slope $\alpha$, one can construct the following network

$$f(w', b', a', x) = \sum_{r=1}^{2m} a'_r \phi_\alpha(w_r'^\top x + b'_r). \tag{39}$$

If we choose $w'_r = w_r = -w'_{r+m}$, $b'_r = b_r = -b'_{r+m}$, $a'_r = \frac{1}{1-\alpha^2} a_r$ and $a'_{r+m} = \frac{\alpha}{1-\alpha^2} a_r$, we have

$$
a'_r \phi_\alpha(w'_r{}^\top x + b'_r) + a'_{r+m} \phi_\alpha(w'_{r+m}{}^\top x + b'_{r+m})
$$
$$
= \frac{1}{1-\alpha^2} a_r \phi_\alpha(w_r^\top x + b_r) - \frac{\alpha^2}{1-\alpha^2} a_r \phi_{\frac{1}{\alpha}}(w_r^\top x + b_r) = a_r \phi(w_r^\top x + b_r), \quad (40)
$$

This immediately suggests that $f(w', b', a', x) = f(w, b, a, x)$.

Since deeper networks, and one with more complex topologies, can be constructed by composing and summing shallower ones, the general claim follows. $\qquad\square$

## D    Experiment Details

For input preprocessing on ImageNet we perform a random crop of size $224 \times 224$ to each image, and apply a random horizontal flip. In all experiments, we applied $L_2$ regularization only to the weights (and not the biases or batch normalization parameters). We selected the $L_2$ constant by grid search from $\{0.00005, 0.00002, 0.0\}$. For networks without batch normalization layers we applied dropout to the penultimate layer, with the dropout rate chosen by grid search from $\{0.2, 0.0\}$. In addition, we used label smoothing (Szegedy et al., 2016) with a value of 0.1.

For each optimizer we used a standard learning rate warm-up scheme which linearly increases the learning rate from 0 to the "initial learning rate" in the first 5 epochs, and then decays the learning rate by a factor of 10 at $4/9$ and $7/9$ of the total epoch budget[5], unless specified otherwise. The initial learning rate was chosen by grid search from $\{1.0, 0.3, 0.1, 0.03, 0.01\}$ for SGD, $\{0.003, 0.001, 0.0003, 0.0001, 0.00003\}$ for K-FAC, and $\{10.0, 3.0, 1.0, 0.3, 0.1\}$ for LARS. For all optimizers we set the momentum constant to 0.9. For K-FAC, we used a fixed damping value of 0.001, and a norm constraint value of 0.001 (see Ba et al. (2017) for a description of this parameter). We also updated the Fisher matrix approximation every iteration, and computed the Fisher inverse every 50 iterations, unless stated otherwise. For LARS, we set the "trust" coefficient to 0.001. For networks with batch normalization layers, we set the decay value for the statistics to 0.9.

For initialization of the weights we used the scale-corrected uniform orthogonal (SUO) distribution (Martens et al., 2021) for all methods/models, unless stated otherwise. For a $m \times k$ matrix (with $k$ being the input dimension), samples from this distribution can be generated by computing $\left(XX^\top\right)^{-1/2} X$, where $X$ is an $m \times k$ matrix with entries sampled independently from $\mathcal{N}(0, 1)$. When $m > k$, we may apply the same procedure but with $k$ and $m$ reversed, and then transpose the result. The resulting matrix is further multiplied by the scaling factor $\max\{\sqrt{m/k}, 1\}$, which will have an effect only when $k \leq m$. For convolutional networks, we initialize only the weights in the center of each filter to non-zero values, which is a technique known as Delta initialization (Balduzzi et al., 2017; Xiao et al., 2018), or Orthogonal Delta initialization when used with orthogonal weights (as we do in this work).

We implemented all methods/models with `JAX` (Bradbury et al., 2018) and `Haiku` (Hennigan et al., 2020). We used the implementation of SGD and LARS from `Optax` (Hessel et al., 2020). We used the JAX implementation of K-FAC available at `https://github.com/deepmind/kfac_jax`.

## E    Additional Experimental Results

### E.1    Empirical c values for finite-width networks

The computation of cosine similarities performed by C maps is only an approximation for finite width networks, and it is natural to ask how large the approximation error is. To answer this question, we compare the theoretical predictions with the empirical simulations on fully-connect networks of different depths and widths. In particular, we use a fixed $\eta = 0.9$ for `TReLU` and we compute the $l$-th "empirical c value" $\hat{c}^l = \frac{x_1^l{}^\top x_2^l}{\|x_1^l\| \|x_2^l\|}$ for each layer index $l$, where $x_1^0$ and $x_2^0$ are random vectors

---

[5]We later found that cosine learning rate annealing (Loshchilov & Hutter, 2016) is slightly better for most settings, but this did not change our conclusions.

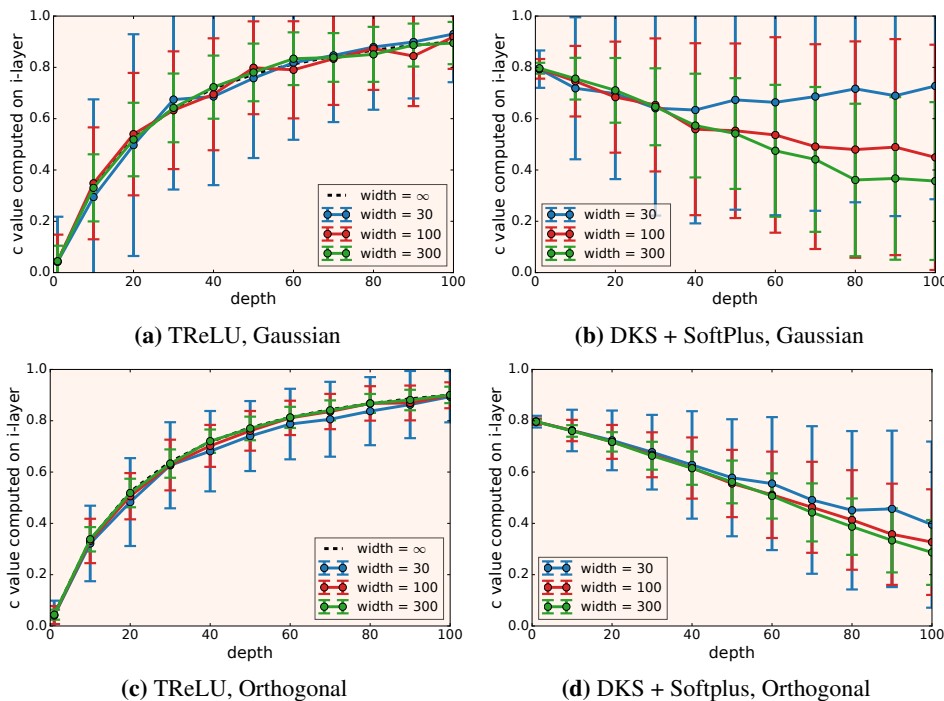

**(a)** TReLU, Gaussian       **(b)** DKS + SoftPlus, Gaussian

**(c)** TReLU, Orthogonal       **(d)** DKS + Softplus, Orthogonal

**Figure 4:** Empirical c values for TAT and DKS, which are averaged over 100 pairs of inputs and 50 different randomly-inialized networks. We include the results for both Gaussian fan-in and Orthogonal initialization. Vertical lines indicate the standard deviation. `TReLU` has smaller kernel approximation error and is robust to Gaussian initialization. For `TReLU`, we also plot the evolution of the c values (black dashed line) as predicted by the C map (which we can compute analytically for `TReLU`).

chosen so that $\|x_1^0\|^2 = \|x_2^0\| = d_0$ and $x_1^{0\top} x_2^0 = 0$ (so that $\hat{c}^0 = 0$). As shown in Figure 4a and 4c, the approximation error is relatively small even for networks with width 30.

We also included the results for networks using DKS (with $\zeta = 10$) and the SoftPlus activation function. Figure 4b and 4d reports empirical c values as a function of layer index $l$, with $x_1^0$ and $x_2^0$ chosen so that $\hat{c}^0 = 0.8$. With Gaussian initialization, the standard deviations are much larger than `TReLU`, and the average values for widths 30 and 100 deviate significantly from the theoretical predictions. (The DKS conditions implies $\mathcal{C}(c) \leq c$ for any $c \in [0, 1]$, which suggests the c value should decrease monotonically.) By comparison, the error seems to be much smaller for orthogonal initialization, which is consistent with the better performance of orthogonal initialization reported by Martens et al. (2021). (By contrast, we show in Appendix E.7 that Gaussian initialization performs on par with orthogonal initialization for `TReLU`.) In addition, we note that the standard deviations increase along with the depth for both Gaussian and orthogonal initializations.

## E.2 RESULTS ON CIFAR-10

In addition to our main results on the ImageNet dataset, we also compared TAT to EOC on CIFAR-10 (Krizhevsky et al., 2009) using vanilla networks derived from a Wide ResNet reference architecture (Zagoruyko & Komodakis, 2016). In particular, we start with a Wide ResNet with a widening factor of 2, and remove all the batch normalization layers and shortcut connections. We trained these networks with the K-FAC optimizer for 200 epochs using a standard piecewise constant learning rate schedule. To be specific, we decay the learning rate by a factor of 10 at 75 and 150 epochs. For K-FAC, we set the damping value to 0.01 and norm constraint value to 0.0001.

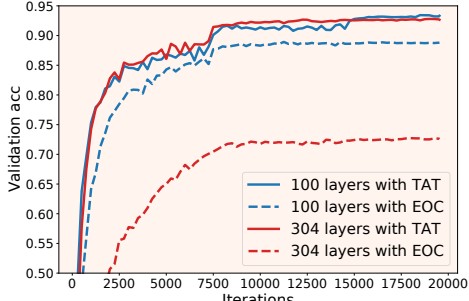

**Figure 5:** CIFAR-10 validation accuracy of ResNets with ReLU activation function initialized using either EOC or TAT (ours).

For data preprocessing we include basic data

augmentations such as random crop and horizontal flip during training. As shown in Figure 5, TAT outperforms EOC significantly. As we increase the depth from 100 to 304, the accuracy of EOC network drops dramatically while the accuracy of the TAT network remains roughly unchanged.

### E.3    REDUCING THE OVERHEAD OF K-FAC

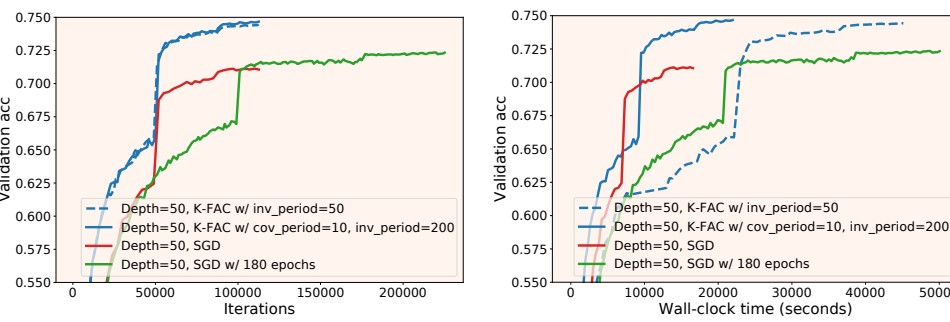

**Figure 6:** Top-1 validation accuracy on ImageNet as a function of number of iterations (**left**) or wall-clock time (**right**) with K-FAC optimizer. One can reduce the computational overhead significantly by updating curvature matrix approximation and its inverse less frequently.

In our main experiments the per-step wall-clock time of K-FAC was roughly $2.5\times$ that of SGD. However, this gap can be decreased significantly by reducing the frequency of the updates of K-FAC's approximate curvature matrix and its inverse. For example, if we update the curvature approximation every 10 steps, and the inverses every 200 steps, the average per-step wall-clock time of K-FAC reduces by half to a mere $1.25\times$ that of SGD. Importantly, as can be seen on Figure 6, this does not appear to significantly affect optimization performance.

### E.4    DISENTANGLING TRAINING AND GENERALIZATION

In our main experiments we only reported validation accuracy on ImageNet, making it hard to tell whether the superior performance of TAT vs EOC is due to improved fitting/optimization speed, or improved generalization. Here, we compare training accuracies of EOC-initialized networks (with ReLU) and networks with `TReLU`, in exactly the same experimental setting as Figure 1. We train each network on ImageNet using K-FAC for 90 epochs. For each setting, we plot the training accuracy for the hyperparameter combination that gave the highest final validation accuracy. As shown in Figure 7, the EOC-initialized networks achieve competitive (if not any better) training accuracy, suggesting that the use of `TReLU` improves the generalization performance and not optimization performance.

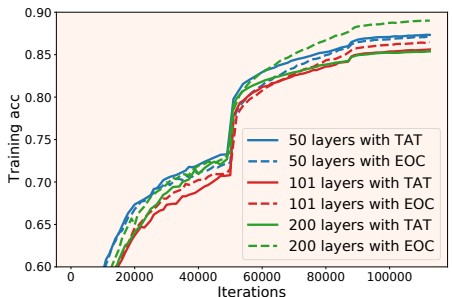

**Figure 7:** ImageNet training accuracy of deep vanilla networks with either EOC-initialized ReLU networks or `TReLU` networks.

### E.5    CLOSING THE REMAINING GAP USING WIDER NETWORKS

In all of our main experiments we used networks derived from standard ResNets (by removing normalization layers and/or shortcut connections). By construction, these have the same layer widths as standard ResNets. A natural question to ask is whether using wider networks would change our results. For example, it's possible that vanilla networks with TAT would benefit more than ResNets from increased width, since higher width would make the kernel approximations more accurate, and could also help compensate for the minor loss of expressive power due to the removal of shortcut connections.

**Table 9:** The effect of increasing width on ImageNet validation accuracy. We use vanilla networks for EOC and TAT (ours).

| Depth | Width | EOC | TAT | ResNets |
|-------|-------|------|------|---------|
| 50    | 1×    | 72.0 | 76.0 | 76.7    |
|       | 2×    | 73.5 | 77.3 | 77.9    |
| 101   | 1×    | 62.4 | 76.5 | 77.9    |
|       | 2×    | 66.5 | 77.6 | 78.6    |

With layers double the width of standard ResNets, it becomes too expensive to store and invert Kronecker factors used in K-FAC. Therefore, we only train these wider networks with SGD. In order to mitigate the slower convergence of SGD for vanilla networks (see Section 5.5), we train them for 360 epochs at a batch size of 512. Note that due to increased overfitting we observed in ResNets after 360 epochs (resulting in lower validation accuracy) we only trained them for 90 epochs. As shown in Table 9, doubling the width does indeed narrow the remaining validation accuracy gap between ResNets and vanilla TAT networks. In particular, the gap goes from 0.7% to 0.6% for depth 50 networks, and from 1.4% to 1% for depth 101 networks.

### E.6 Comparison with PReLU on Rescaled ResNets

In Table 6 of the main text we compare PReLU and TReLU on deep vanilla networks. Here we extend this comparison to rescaled ResNets with a shortcut weight of $w = 0.8$. For PReLU, we again include two different initializations: one with 0 negative slope (effectively ReLU), and another with $0.25$ negative slope (which was used in He et al. (2015)). We report the full results in Table 10. For all settings, TAT outperforms PReLU by a large margin, suggesting that a better-initialized negative slope is crucial for both rescaled ResNets and deep vanilla networks.

**Table 10:** Comparison with PReLU with rescaled ResNets ($w = 0.8$).

| Depth | Optimizer | TReLU | PReLU$_{0.0}$ | PReLU$_{0.25}$ |
|---|---|---|---|---|
| 50 | K-FAC | 76.4 | 75.7 | 73.6 |
| | SGD | 76.0 | 71.5 | 71.5 |
| 101 | K-FAC | 77.8 | 76.4 | 76.8 |
| | SGD | 77.3 | 73.1 | 73.4 |

### E.7 Comparison of different initializations

In all of our experiments we use the Orthogonal Delta initialization introduced by Balduzzi et al. (2017) and Xiao et al. (2018). This is because it's technically required in order to apply the extended Q/C map analysis of Martens et al. (2021) (which underlies DKS and TAT) to convolutional networks, and because it is generally thought to be beneficial. In this subsection we examine this choice more closely by comparing it to a traditional Gaussian fan-in initialization (with $\sigma_w^2 = 2$ for ReLUs). We consider standard ResNets and

**Table 11:** Comparison of Orthogonal Delta and Gaussian fan-in initialization.

| Depth | Optimizer | Init | ResNet | EOC | TAT |
|---|---|---|---|---|---|
| 50 | K-FAC | Orth Delta | 76.4 | 72.6 | 74.6 |
| | | Gaussian | 76.5 | 72.5 | 74.8 |
| | SGD | Orth Delta | 76.3 | 63.7 | 71.0 |
| | | Gaussian | 76.6 | 63.1 | 68.7 |
| 101 | K-FAC | Orth Delta | 77.8 | 71.8 | 74.2 |
| | | Gaussian | 77.8 | 72.3 | 74.1 |
| | SGD | Orth Delta | 77.9 | 41.6 | 70.0 |
| | | Gaussian | 78.0 | 41.1 | 68.7 |

deep vanilla networks using either EOC (with ReLUs) or TAT with (with LReLU). Surprisingly, it turns out that the Orthogonal Delta initialization does not have any clear advantage over the Gaussian fan-in approach, at least in terms of validation accuracy after 90 epochs.

