# OpenReview forum: "Deep Learning without Shortcuts: Shaping the Kernel with Tailored Rectifiers"
_ICLR.cc/2022/Conference — ICLR 2022 Poster_

### Official Review · Reviewer_HqiL · 2021-11-02

**Correctness:** 3
**Technical Novelty And Significance:** 3
**Empirical Novelty And Significance:** 3
**Recommendation:** 6
**Confidence:** 4

**Main Review:**

## Pros

1. This paper is well-organized and easy to follow.  The main contribution of this work is to adapt DKS to models with ReLU-family activations. The authors have explored the reason why DKS is not fully compatible with ReLU activations, then providing reasonable solutions by changing some Q/C Maps conditions. The proposed methods can be easily applied to activation layers and only involve a few extra computations.

2. Effectiveness of  the proposed TReLU has been demonstrated by a series of ablation studies. With K-FAC optimizer, the TReLU ResNet can maintain performance without any residual connections and normalization in some cases.

## Cons

1. The problem of removing residual connections on NN is interesting, and this paper has provided a good solution. However, existing methods seem not to be so bad as claimed in the paper. As discussed in the paper, the main drawback of DKS is incompatibility with ReLU-family activations. But as shown in Table.7, DKS+LReLU+K-FAC+ResNet50 seems to slightly outperform TAT counterparts, which means the mentioned incompatibility of DKS can be alleviated by using other methods like K-FAC optimizer and thus making TAT not so significant. Anyway, I thought DKS is a very recent work compared to this paper, and the proposed TAT can also yield comparable results.
Therefore,  I still lean to accept this work.

**Summary Of The Paper:**

This paper mainly discusses the training of neural networks without residual connections. To close the gap between residual-free and regular models, an activation transformation technique named "Tailored Activation Transformation (TAT)" is introduced.  Compared to the state-of-art method DKS,  the proposed TAT can yield better results for models using ReLU-family activation. Overall the motivation is well-discussed and sufficient ablation studies are provided.

**Summary Of The Review:**

The paper provides sufficient discussion and experiments for improving DKS. The proposed approaches yield good results on models without residuals and normalizations.  However, when compared to existing DKS methods, the improvements seem to be somewhat marginal, thus restricting the significance of this paper.

---

> ### Author Response · Authors · 2021-11-18
> **Response to Reviewer HqiL**
>
> Thank you for the review and comments. We address your concerns below.
>
> **Comparison between TAT (TReLU) and DKS**
> - We believe TReLU is still slightly better than DKS overall according to table 7. In addition, the technique for applying DKS to LReLU activations is proposed in our paper. In the original DKS paper, the authors only used ReLU and had to drop one of the Q/C map conditions, resulting in significantly worse performance.
> - We want to emphasize that all previous works on training vanilla neural networks (including the original DKS paper) have failed to show competitive validation accuracy on large-scale problems. Our paper is the **first work** that achieves competitive performance on ImageNet with deep vanilla networks, and we believe this alone is a significant contribution. It conveys the important message that shortcut connections are not necessary to achieve good results with deep networks.
> - We agree that the improvement over DKS is relatively small with careful tuning (as done in our paper), but the original DKS paper didn’t show such competitive results. Indeed, the results in the original paper fell substantially short of ResNets in terms of generalization performance.
> - In addition, TReLU enjoys better theoretical properties:
>     - First, it has an identity Q map, and a C map which is independent of the q values. As we argue in the paper, this implies lower kernel approximation error in realistically wide networks compared to DKS, which might have some **practical implications**. In support of this, we added some results in Appendix E.1 (updated version), which show that TReLU does indeed have a smaller kernel approximation error in practice compared to DKS with Softplus units. In addition, the variance of c values (in the case of DKS) becomes larger for deeper layers.  We emphasize that the C map computation of DKS is very inaccurate for Gaussian initialized networks of small widths (see Figure 4(b)). It is likely that TReLU would perform much better than DKS in the case of small width and Gaussian initialization.
>     - Second, we showed in Proposition 2 and Figure 2 that the C map of TReLU converges to a nice function quickly with a fixed eta. This indicates we have control of the full C map by only tuning eta. For DKS, we can only control how close it is to the identity function.
>
>
> Please kindly consider raising the score if our response addresses your concerns. If not, please let us know, we are happy to answer more questions.

---

### Official Review · Reviewer_TjGY · 2021-11-02

**Correctness:** 4
**Technical Novelty And Significance:** 3
**Empirical Novelty And Significance:** 2
**Recommendation:** 6
**Confidence:** 3

**Main Review:**

This paper makes solid theoretical contributions on analyzing the initial conditioning of feedforward networks without shortcuts. Most of the analysis are derived directly from the concurrent work (DKS). It provides insights for proper initialization and configuring the activations for feedforward networks to achieve rapid training speed, improves our understanding of neural networks and has the potential to impact future design choices of them. Although the main point of the paper is to remove shortcuts, the analysis can also be applied to rescale the shortcuts of ResNets and achieve even better results than standard ResNets (with BN) when BN is not used (Table 4).

I feel the only weaknesses of the paper are the clarity and the significance of the results. Limited by the pages, many important details are either abbreviated or require reading the 172-page DKS paper. The results of shortcut-free networks are still slightly worse than ResNets, and the merits of removing shortcuts do not seem to offset the drawbacks. A minor concern is the current analysis cannot be applied to Batch Normalizations and attention mechanisms, but I believe the theoretical contribution is enough.

In addition, I have following questions or suggestions:

1. For each experimental result, how many different random seeds are used, and what are the standard errors of these results?

2. Is training using K-FAC really faster than SGD? Most tables show better results achieved with K-FAC than SGD under the same number of epochs, and it has been shown in Figure 5 that K-FAC can be made 2x as fast without losing accuracy, but the wall-clock time comparison with SGD is not shown. How is the wall-clock time like for the 90-epoch K-FAC and 180-epoch SGD?

3. The paper did not clearly specify how the weights are initialized. It is mentioned in Section D.6 in the appendix, but it was not clearly specified in Section 2.1. It would be better to include some form of pseudo code to specify the weight initialization and setting the parameters of activations. It is also not clearly described in Section 4.2 that the smooth activations are transformed.

4. The method itself introduces extra hyperparameters that require additional hyperparameter search, such as $\eta$. This increases the cost of hyperparamter search. Could there be some ablation studies on the impact of $\eta$?

5. How is the result like for even deeper networks, e.g., ResNet-152?


**Summary Of The Paper:**

Based on the analysis in the concurrent work (DKS), this work develops a new set of conditions for the Q/C maps to set the non-trainable parameters for the activations (Leaky ReLU and transformed smooth activations), which enables training deep feedforward networks at comparable speed as ResNets on ImageNet, and achieves significantly better test accuracy than those obtained with EOC.

**Summary Of The Review:**

This paper makes solid theoretical contributions on analyzing the initial conditioning of feedforward networks without shortcuts. The analysis can even be used to rescale the shortcuts of BN-free ResNets and achieve even better results than standard ResNets (with BN). However, the method also introduces an extra hyperparameter $\eta$, and results of shortcut-free networks are still slightly worse than ResNets. The clarity of the paper also needs improvements.

---

> ### Author Response · Authors · 2021-11-18
> **Response to Reviewer TjGY**
>
> Thank you for your detailed comments. We address your comments below.
>
> **Limited by the pages, many important details are either abbreviated or require reading the 172-page DKS paper.**
> - We have updated the paper to include many additional details in the appendix. If there is anything you still find vague or unclear please let us know.
>
> **The results of shortcut-free networks are still slightly worse than ResNets, and the merits of removing shortcuts do not seem to offset the drawbacks.**
> - In terms of the merits of removing shortcuts, we want to emphasize that training vanilla deep neural networks is an interesting research problem in its own right, and finding a solution could open the path to discovering entirely new model architectures. In particular, our method may enable new architectures to be trained for which existing techniques, such as shortcuts and normalization layers, are insufficient.
> - In terms of the comparison with standard ResNets, we note the gap is relatively small (0.6%) for ResNet50 if we train for 180 epochs (see table 4). We admit that it still takes slightly longer for our networks to converge, but we believe the training speed could be improved in the future to close the gap. For deeper networks, we believe the gap is due to the ensemble-like behavior of deep ResNets. At high depths, one can think of a ResNet as an ensemble of shallower models. It’s possible that with some sort of ensembling techniques for vanilla networks, we can achieve exactly the same or even better performance, but we will leave this to future work. That being said, this is the first time one can achieve such competitive results on ImageNet with vanilla networks.
>
> **The current analysis cannot be applied to Batch Normalizations and attention mechanisms**
> - Yes, we agree that our current analysis is not applicable for batch normalizations and attention mechanisms. In terms of batch norm, that is exactly something we want to remove. Many recent works are trying hard to get rid of batch normalization, as batch normalization has many undesirable properties stemming from its dependence on the batch size and interactions between examples. To our knowledge, batch norm has long been a headache for both practitioners and theorists and we believe batch norm will eventually be discarded in our deep learning pipeline. So we don’t see this as a weakness for our paper. As for attention mechanisms, the extension of our theory to the self-attention mechanism is nontrivial and we leave this to future work.
>
> **Random Seeds**
> - As for random seeds, we only used 1 random seed as it is extremely expensive to run our experiments with multiple seeds (for each number reported in the paper, it already requires training ~30 models on ImageNet due to hyperparameter tunning). We understand that it could raise the concern of reproducibility, but in practice, the results on ImageNet classification are quite robust (see e.g. Table 2 of [4]) and many papers (see e.g., [1, 2, 3]) only report results with one seed. We did try running with 3 seeds in a few experiments, and the standard deviation for the validation accuracy was typically smaller than 0.1.
>     - [1] Patches Are All You Need?
>     - [2] Batch Normalization Biases Residual Blocks Towards the Identity Function in Deep Networks. NeurIPS 2020
>     - [3] RepVGG: Making VGG-style ConvNets Great Again. CVPR 2021.
>     - [4] An Image is Worth 16x16 Words: Transformers for Image Recognition at Scale. ICLR 2021.
>
> **Training speed of K-FAC**
> - We have some discussions in Appendix E.3 about the speed of K-FAC vs SGD. By updating the covariance and inverse less frequently, the wall-clock time per-iteration is only 1.25x that of SGD. In the updated version, we added the curves of SGD for wall-clock time comparison (see Figure 6).
>
> **Details about weight initialization and activation transformation**
> - In terms of weight initialization, we added some details at the beginning of the experiments section and gave detailed formulas in Appendix D. We also had some ablation studies for weight initialization in Appendix E.7. In the updated version, we added pseudocode for the activation transformations in Appendix B. A Jax implementation was also provided in Appendix F (updated version).
>
> **Sensitivity of $\eta$**
> - In terms of the sensitivity of eta, for TReLU we only tuned over {0.9, 0.95}. And more importantly, simply choosing 0.9 for all the settings won’t hurt the performance too much. The performance of eta = 0.9 is quite close to eta = 0.95. The reason why we did this grid search was that we wanted to make sure our comparison with DKS, for which we also searched over hyperparameters, was as fair as possible.
>
> **Results for deeper networks**
> - For deeper networks, it is a bit too expensive to include results for all the settings. But we did report results for a 200 layer network in Figure 1. In addition, we have results of 300-layer networks on CIFAR (see Appendix E.2).

---

### Official Review · Reviewer_vixN · 2021-11-03

**Correctness:** 4
**Technical Novelty And Significance:** 3
**Empirical Novelty And Significance:** 3
**Recommendation:** 6
**Confidence:** 2

**Main Review:**

Strength: (1) The paper is well-organized --- although the technical backgrounds are complicated, the authors try to introduce them in reasonably short sections. (2) The paper succeeds in increasing the performance of deep vanilla networks with ReLU-family activation, matching the performance of deep residual networks.

Weakness: (1) The current draft is unfriendly to practitioners. Maybe the authors would like to include an algorithm box at the beginning to explain the proposed method (without understanding how to derive it). (2) Although the proposed method supports ReLU-family activation (TReLU), the performance gain is very marginal compared to smooth activations with DKS (at most 0.3%, Table 7). Thus, it raises the concern whether ReLU-family activation is needed. Is there any tradeoff using ReLU-family activation verse using smooth activation?

**Summary Of The Paper:**

The paper extends the Deep Kernel Shaping (DKS) method to Tailored Activation Transformation (TAT). As a result, deep vanilla neural networks with ReLU-family activations match the performance of deep residual networks.

**Summary Of The Review:**

Although the authors have well-organized the paper, the current draft is hard to follow due to its technical complicity. Moreover, given the marginal improvement compared to DKS, a more extensive discussion on smooth and ReLU activations is desirable.

---

> ### Author Response · Authors · 2021-11-18
> **Response to Reviewer vixN**
>
> Thank you for the comments and feedback. We address your two concerns below.
>
> **The current draft is unfriendly to practitioners**
> - To make the paper more accessible, we added pseudocode in Appendix B. Due to the page constraint, we won’t be able to put it in the main paper. For detailed implementation, we provided the Jax code in Appendix F. We also plan to open source a cleaned-up version of the code in the next month or two.
>
> **Comparison between TAT and DKS, ReLU-like activation, and smooth activations**
> - We want to emphasize that all previous works on training vanilla neural networks (including the original DKS paper) have failed to show competitive validation accuracy on large-scale problems. Our paper is the **first** to do this (with DKS, TAT or any other method), and we believe this alone is a significant contribution. It conveys the important message that residual connections are not necessary to achieve good results with deep networks.
> - We agree that the improvement over DKS is relatively small with careful tuning (as done in our paper), but **the original DKS paper didn’t show such competitive results**. Indeed, the results in the original paper fell substantially short of ResNets in terms of generalization performance.
> - In addition, TReLU enjoys better theoretical properties:
>     - First, it has an identity Q map, and a C map which is independent of the q values. As we argue in the paper, this implies lower kernel approximation error in realistically wide networks compared to DKS, which might have some practical implications. In support of this, we added some results in Appendix E.1 (updated version), which show that TReLU does indeed have a smaller kernel approximation error in practice compared to DKS with Softplus units. In addition, the variance of c values (in the case of DKS) becomes larger for deeper layers.
>     - Second, we showed in Proposition 2 and Figure 2 that the C map of TReLU converges to a nice function quickly with a fixed eta. This indicates we have control of the full C map by only tuning eta. For DKS, we can only control how close it is to the identity function.
> - Simply showing that DKS or TAT is compatible with positive homogenous functions itself is important. Many practitioners might like to use ReLU or LReLU as ReLU-like activations are computationally simple.
> - One possible explanation for the small gap between TReLU and DKS + Softplus is that Softplus itself is indeed similar to ReLU, so its Q/C maps might be “better” compared to other smooth activations (such as Tanh), though the DKS paper doesn’t explicitly enforce that. We note the performance of DKS + Tanh is indeed worse. Nevertheless, it is also possible that in some other settings, TReLU would perform much better. That being said, it is hard to predict the future impact.
>
>
> Please kindly consider raising the score if our response addresses your concerns. If not, please let us know, we are happy to answer more questions.

---

### Official Review · Reviewer_8mde · 2021-11-23

**Correctness:** 4
**Technical Novelty And Significance:** 3
**Empirical Novelty And Significance:** 3
**Recommendation:** 6
**Confidence:** 3

**Main Review:**

Strong points:

1. A study of an interesting problem with rigorous theoretical analysis.

2. Both theoretical extension and empirical improvement.

3. Solves the ReLU incompatibility issue of existing DKS work with new Q/C map conditions and LReLu, which also looks nicer than the the DKS's model class-preserving transformation.

Weak points:

I only have one major concern. The claims that DHS is not compatible are a bit too strong, at least the experiments (LReLu+DKS) did not show that way. At the end of Section 3, it says "However, DKS is not fully compatible with ReLUs,", which needs an exact explanation with the DKS condition.  DHS is compatible with LReLUs, and works pretty well, as in Table 7. So overall,  I feel some of the claims frequently mentioned throughout the paper are a little bit exaggerating. DHK, in their paper, claims quite differently, "small decrease in generalization performance" for removing BN, and "similar results" for removing skip connections.

The other claim is ** Using TAT, we demonstrate for the first time that a 50-layer vanilla deep network can nearly match the validation
accuracy of its ResNet counterpart when trained on Imagenet.**. What "for the first time, ..., nearly..." means? I can tell, from Table 7, that  DKS is also very close, even with SoftPlus if you insist LReLU+DKS is your contribution.



**Summary Of The Paper:**

This paper studied the problem of DNN training and generalization in vanilla architecture (without BN and Skip Connections in ResNets). It follows NTK theory and the approach of applying certain transformations to the activation functions. This work improves an existing work DKS, and solves its incompatibility to ReLU activations ("Leaky" ReLUs). This work introduces the necessary modifications to the Q/C map conditions for using Leaky ReLUs and shows empirical improvement over DKS or an easier method EOC on ImageNet.

**Summary Of The Review:**

I would suggest the authors be careful about their claims. It appears to me that some extension from DKS motivated by the similarity between the model class-preserving transformation and Leaky ReLU is nice, but is not world-changing.

I appreciate the detailed derivations.

My overall rating is 6.

---

> ### Author Response · Authors · 2021-11-23
> **Response to Reviewer 8mde**
>
> Thank you for the review and comments. We address your concern below.
>
> **The claim that DKS is not compatible with ReLUs is too strong**
>
> - In the DKS paper, the authors proposed four conditions on the Q/C maps (see table 1 of our paper). To achieve these four conditions, they used a particular transformation of $\hat{\phi}(x) = \gamma (\phi(\alpha x + \beta) + \delta)$ for the activation function. However, it only has three degrees of freedom for ReLU. So in the case of ReLU, they had to drop one condition (we explained this in the first paragraph of section 4). This results in a significant performance drop in both training and test sets, as shown in the figures on pages 101 and 102 (of the DKS paper). The introduction of LReLU and how to make it work with DKS (using a slightly different transformation $\tilde{\phi}(x) = \gamma (\phi_\alpha(x + \beta) + \delta)$, where $\phi_\alpha(x)$ is LReLU with negative slope $\alpha$) are the contributions of our paper.
> - In the DKS paper, the final test accuracy for the best activation function is around 65% on ImageNet (see page 102). They used different learning rate schedules and removed all regularizations, and the gap between ResNets and DKS networks is significant (~7%). In our paper, we can achieve 76% accuracy with TReLU if we train 180 epochs, which is only 0.6% lower compared to ResNets of the same depth. In this sense, we believe it is fair to claim that “for the first time …, nearly …”.

---

### Public Comment · ~Yao_Lu14 · 2021-11-10
**Related Work**

Interesting work! But I think our work [1] should be cited for two relevant points.

1. Random orthogonal weight matrices under Haar measure. While the use of orthogonal matrices for initialization has been proposed before, our work first exploits random orthogonal matrices under Haar measure for the gradient behaviors of neural networks.

2. Transformation of activation functions. Our work also discusses this topic for improving neural network trainability using a similar scheme.

[1] Bidirectionally Self-Normalizing Neural Networks
Yao Lu, Stephen Gould, Thalaiyasingam Ajanthan, 2020.
https://arxiv.org/abs/2006.12169

---

### Decision · Program_Chairs · 2022-01-20

**Decision:**

Accept (Poster)

**Comment:**

This paper seeks to find an answer to some quite interesting research question: can deep vanilla networks without skip connections or normalization layers be trained as fast and accurately as ResNets? In this regard, the authors extend Deep Kernel Shaping and show that a vanilla network with leaky RELU-family activations can match the performance of a deep residual network.

Four reviewers unanimously suggested acceptance of the paper. There were concerns about the clarity or marginal performance improvement. However, they all including myself agree: achieving the competitive performance with the vanilla deep model itself can be seen as a big contribution and the clarity has been improved to some extent through revision.